# A Stable Whitening Optimizer for Efficient Neural Network Training

**Kevin Frans**
UC Berkeley
kvfrans@berkeley.edu

**Sergey Levine**
UC Berkeley

**Pieter Abbeel**
UC Berkeley

## Abstract

In this work, we take an experimentally grounded look at neural network optimization. Building on the Shampoo family of algorithms, we identify and alleviate three key issues, resulting in the proposed SPlus method. First, we find that naive Shampoo is prone to divergence when matrix-inverses are cached for long periods. We introduce an alternate bounded update combining a historical eigenbasis with instantaneous normalization, resulting in across-the-board stability and significantly lower computational requirements. Second, we adapt a shape-aware scaling to enable learning rate transfer across network width. Third, we find that high learning rates result in large parameter noise, and propose a simple iterate-averaging scheme which unblocks faster learning. To properly confirm these findings, we introduce a pointed Transformer training benchmark, considering three objectives (language modelling, image classification, and diffusion modelling) across different stages of training. On average, SPlus is able to reach the validation performance of Adam within $44 - 58\%$ of the gradient steps and $62 - 83\%$ of the wallclock time.

## 1 Introduction

The backbone of modern deep learning is stochastic gradient descent – a method which, while particularly effective in practice, is time-consuming by its iterative nature. As large neural networks are scaled up to billions of parameters [1, 8, 38], and are trained on datasets of similar scale, it becomes increasingly important that each gradient update is efficient in making learning progress.

A core optimization strategy involves adapting updates to second-order curvature [3, 24], allowing for faster learning progress in certain directions while preventing divergence in others. However, practical challenges arise with regards to large neural network training. For example, it is computationally intractable to calculate the Hessian of any reasonably-sized network [25, 33], and even further intractable to invert this matrix. Numerical stability is also a concern, and complex methods often require additional hyperparameter tuning and regularization [2, 32].

In this work, we take a cautiously empirical view on neural network optimization. We carefully design an evaluation suite that is well-aligned with common uses today. We consider the optimization of a standard Transformer model [52] on *three distinct training settings* – autoregressive language modelling [42], diffusion modelling [16, 37], and image classification [10]. We further consider the performance of optimizers at various stages of training (e.g. starting from checkpoints near the start, middle, and end), to avoid biasing towards early-stage performance gains. To our knowledge, this is currently the widest-scope comparison of adaptive optimizers on Transformer training.

From findings in this setting, we develop a method we refer to as SPlus. SPlus builds on the Shampoo family of algorithms [15], which can broadly be seen as approximating a *whitening metric* from historical gradients, and performing steepest descent along this metric. We identify and address three issues with the naive Shampoo update. First, we find that Shampoo often diverges at high learning

39th Conference on Neural Information Processing Systems (NeurIPS 2025).

rates or under less-frequent inversion rates. To fix this issue, we introduce an alternative bounded update combining the historical eigenbasis with instantaneous normalization, resulting in across-the-board stability even with significantly lower-frequency inversions. Second, we find that Shampoo update magnitudes are not properly scaled with relation to network width. We adopt methodology derived for SGD and Adam [55] to our setting, enabling easy hyperparameter tuning via learning rate transfer across network widths. Third, we find that higher learning rates in whitening-based optimizers result in significant parameter noise. We propose a simple iterate-averaging scheme to alleviate this issue, unblocking fast learning with a much lower degradation in performance.

In experiments, SPlus achieves the fastest loss decrease among a wide suite of previous optimizers. SPlus matches Adam within $\sim 44 - 58\%$ of the gradient steps, and within $\sim 62 - 83\%$ of the wallclock time. SPlus does not introduce any critical hyperparameters to tune, and does not require additional forward/backward passes. We hope that this effectiveness and simplicity enables the community to easily adopt SPlus to their existing training settings.

We provide the full code to replicate experiments at github.com/kvfrans/splus. The repo also contains single-file implementations of SPlus in JAX and Pytorch, along with basic reccomendations for usage.

## 2   Related Work

**Second-order optimization of neural networks.** Second-order methods, which we broadly define as methods which further modify the first-order gradient, can be largely categorized among two axes – the second-order metric used, and the way in which the metric is approximated and applied. A common metric choice is the Hessian, which in Newton's method can be directly applied by multiplying the inverse Hessian and the gradient. Neural-network specific methods have approximated the Hessian via conjugate-gradient methods [33], iterative fitting [25, 41], or a diagonal approximation [24, 27]. An alternate metric is the Fisher, motivated by natural gradient methods [3, 49]. Closest related to our work is the use of the "empirical" Fisher metric [22], i.e. an uncentered covariance matrix of gradients. The most well-tested of these are the diagonal Adam and its variants [11, 20, 28, 46].

**Kronecker factorized metrics.** Our work utilizes a Kronecker factorization of the empirical Fisher, building on the techniques introduced in K-FAC [32], and more directly Shampoo [15]. Shampoo has been extended to utilize distributed inversion and exponential averaging [47], and/or perform inversion via Newton iteration [4]. Closest related to our work is SOAP [53], which similarly utilizes an eigen-decomposition of the factor matrices, however, SOAP maintains an additional set of second-moments per parameter, whereas in our work this is not neccessary.

**Non-Euclidean gradient descent.** We build on previous works which have studied gradient descent over non-Euclidean distance metrics. In particular, such a framing may manifest not as linear transformations of the gradient, but also methods which explicitly parameterize learning rate [23, 55], or instantaneously transform the gradient via sign descent [5]. Newton-Shulz orthogonalization of gradients [18, 29] can also be motivated as maximum spectral descent.

**Iterate averaging.** Whitening-based optimizers have poor convergence properties due to the uniform size of each update, and instead typically rely on learning-rate decay. An alternative strategy is to average fast-moving snapshots of parameters [17, 35, 39, 40, 44], which has been shown effective in approximating learning-rate decay [9] and is often used in image generation models [19, 57].

## 3   Background and Preliminaries

Gradient descent methods can be seen as following the steepest descent direction under some metric. Naive gradient descent implicitly assumes a Euclidean distance metric over parameters, in which case the update vector is simply the scaled gradient $g = \nabla_\theta L(\theta, x)$ itself:

$$u = \underset{\Delta\theta}{\arg\min} \; \underbrace{g^T \Delta\theta}_{\text{Improvement}} + \underbrace{(\alpha/2)||\Delta\theta||^2}_{\text{Distance Penalty}} \;\; = \;\; \alpha g. \tag{1}$$

However, it is often helpful to impose other metrics. For example, certain parameters may be more sensitive to second-order changes, and thus should be assigned a larger penalty. We can generally express distance using a Riemannian metric tensor $M$, which is a symmetric positive-definite matrix of shape $\mathbb{R}^{dim(\theta) \times dim(\theta)}$. Under $M$, the distance of an update can be expressed as the matrix product:

$$||\Delta\theta||_M^2 = \Delta\theta^T M \Delta\theta. \tag{2}$$

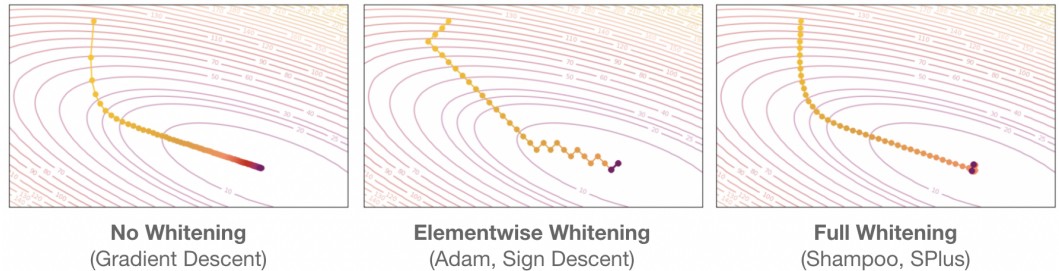

**Figure 1: Whitening normalizes gradients to have uniform magnitude along each axis of descent.**
This decouples the updates from gradient magnitude. Elementwise whitening imposes an independent
axis per dimension, whereas full whitening uses the axes that maximally explain gradient covariance.

Gradient descent can now be performed using $M$ as the distance metric. The solution then becomes:

$$u = \arg\min_{\Delta\theta} \underbrace{g^T\Delta\theta}_{\text{Improvement}} + \underbrace{(1/2)\Delta\theta^T M \Delta\theta}_{\text{Distance Penalty}} = M^{-1}g. \qquad (3)$$

which, in the case of Euclidean distance represented by the identity metric, reduces to Equation (1).

**Whitening metric.** Empirically, the *whitening metric* [56] has proven to be a reliable choice for
neural network optimization. The whitening metric is the square-root of the uncentered covariance:

$$M = \mathbb{E}_x \left[ \nabla_\theta L(\theta, x) \nabla_\theta L(\theta, x)^T \right]^{1/2} = \mathbb{E}_x \left[ gg^T \right]^{1/2} \qquad (4)$$

and is guaranteed to be positive definite. The whitening metric can be calculated via empirical gradi-
ents and does not require additional forward/backward passes. Notably, Adam [20] performs descent
on a diagonal approximation of the whitening metric, assuming each parameter as independent.

**What about the Hessian or the Fisher?** While potentially effective, these metrics are often expensive
to compute as they require additional information outside the standard gradient. We refer to detailed
discussions in [21, 31, 34, 49], as well as a brief overview in Appendix A.4.

**Approximations for neural network learning.** To make storing and inverting the metric amenable
for large neural networks, it is common to assume a per-layer blockwise approximation [15, 32]. In
this way, the full whitening metric can be represented as a set of smaller block matrices, one per layer,
which can each be independently inverted.

To further reduce memory and computation, each block can be further approximated by a Kronecker
product of two smaller matrices:

$$M^{mn,mn} = A^{m,m} \otimes B^{n,n} = \begin{bmatrix} a_{11}B & a_{12}B & \cdots & a_{1m}B \\ \vdots & \vdots & \ddots & \vdots \\ a_{m1}B & a_{m2}B & \cdots & a_{mm}B \end{bmatrix}. \qquad (5)$$

Kronecker products have a useful property that the inverse (at any power) of a Kronecker product is
equivalent to the Kronecker product of the inverse factors:

$$\text{if} \quad M = A \otimes B, \quad \text{then} \quad M^{-1/2} = (A^{-1/2} \otimes B^{-1/2}). \qquad (6)$$

Additionally, multiplication by a Kronecker product can be performed without explicitly forming
the full product matrix. Consider the flattened gradient and update vectors $g, u \in \mathbb{R}^{mn}$ and their
corresponding matrix forms $G, U \in \mathbb{R}^{m \times n}$. The following operations are identical:

$$u = (A \otimes B)^{-1/2}g \quad \leftrightarrow \quad U = A^{-1/2}GB^{-1/2} \qquad (7)$$

In Shampoo [15] (Algorithm 1), the above techniques are utilized to derive an efficient update. The
factor matrices of the whitening metric can be directly calculated from the matrix-shaped gradients:

$$M = E[gg^T]^{-1/2} \quad \approx \quad (L \otimes R) \quad \text{where} \quad L = E[GG^T]^{-1/4} \qquad R = E[G^TG]^{-1/4} \qquad (8)$$

after which the update can be calculated as:

$$u = M^{-1}g \quad \approx \quad U = L^{-1/4}GR^{-1/4}. \qquad (9)$$

In practice, matrix inversion is slow, so the above inversion is performed only every $N$ steps, and the
results are cached until recomputation. This caching can result in unstable training (Figure 2), and is
one of the issues we will discuss in the following sections.

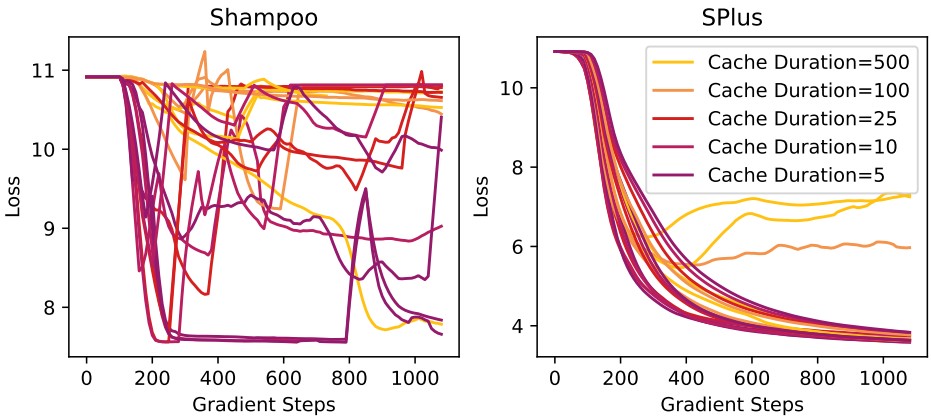

Figure 2: **Shampoo is prone to divergence, but SPlus remains stable under the same settings.**
Plotted above are loss curves on language modelling, sweeping over learning rate between
$(0.0001, 0.000215, 0.000464, 0.001)$ and cache duration between $(5, 10, 25, 100, 500)$. SPlus is significantly more robust to hyperparameters than Shampoo. *This robustness is crucial for improving practical training speed* – in our setting. Shampoo diverges when caching for $> 100$ gradient steps while SPlus remains stable, enabling a faster wall-clock performance than Adam.

## 4  SPlus: A Stable Whitening Optimizer

Our main contribution is SPlus, an efficient optimizer which builds upon Shampoo to stabilize training and reduce overall gradient-step and wall-clock time. In developing SPlus, we take a fundamentally empirical and experimental approach – we identify three core shortcomings of the naive Shampoo method, examine their causes, and propose a series of nuanced improvements to alleviate these issues. In aggregate, these changes lead to a significant improvement in reliability and training speed.

### 4.1   To reliably prevent divergence, utilize instant-sign normalization

As shown in Figure 2, we find that naive Shampoo is prone to divergence. When examining a range of learning rates and matrix-inversion frequencies, we find that Shampoo diverges in $> 50\%$ of the trials in our setting. Notably, Shampoo regularly diverges when matrix-inversion is cached for over 25 gradient steps.

We hypothesize that the interaction between *cached* matrix-inverses and incoming gradients is the cause of frequent divergence. To provide intuition on this behavior, we can rewrite the square-root matrix inverse in terms of its eigen-decomposition[1]:

$$U_{Shampoo} = (Q_L \Lambda_L^{-1/4} Q_L^T) \, G \, (Q_R \Lambda_R^{-1/4} Q_R^T) \tag{10}$$

where eigenvectors $Q$ are orthonormal, and eigenvalues $\Lambda$ are diagonal.

In the above decomposition, the eigenvectors can be understood as basis directions that maximally explain the covariance between gradients. Each eigen*value* represents the historical squared magnitude of gradients along each basis. The Shampoo update normalizes incoming gradients by their respective historical magnitudes along each basis.

The risk is when incoming gradients align with a tail-end basis direction (which has a small historical magnitude), in which case the update can diverge. This risk is especially prominent when the cached matrix-inverse is stale, as incoming gradients may no longer align with the historical distribution.

To alleviate this risk, we instead propose a normalization scheme that does not rely on historical magnitudes at all. Sign-normalization has been previously studied as the equivalence of Adam without a running average, accomplishing a similar normalizing behavior [5, 6]. We therefore opt to ignore historical magnitudes, and instead perform normalization instantaneously via the 'sign'

---

[1]Eigen-decomposition is commonly used under the hood for calculating symmetric matrix inverses, as matrix powers share the same eigenbasis, and the diagonal term is easily raised to any power.

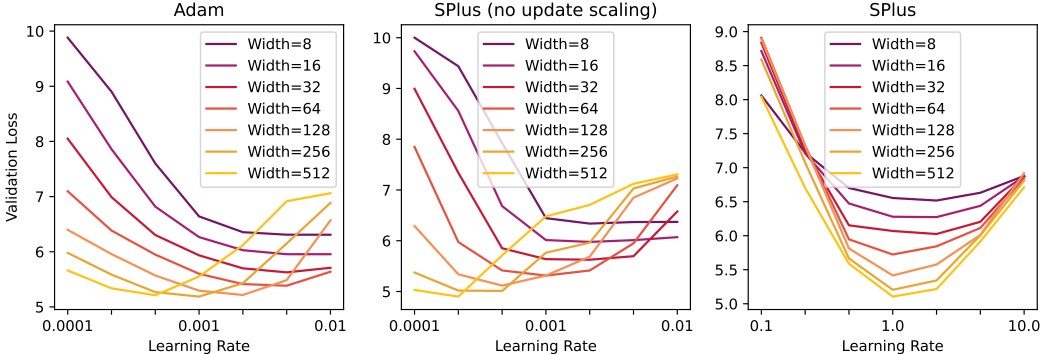

Figure 3: **Optimal learning rates for SPlus transfer across network widths.** This is achieved by normalizing per-layer update magnitudes by constant shape-dependent factor. Notably, this learning rate transfer does not hold by default for Adam or Shampoo.

function. The eigenbasis remains as the historical eigenbasis. We refer to this update as *instant-sign normalization*:

$$U = Q_L sign(Q_L^T G Q_R) Q_R^T. \tag{11}$$

Instant-sign normalization has a hard bound preventing divergence. As $Q_L/Q_R$ are orthonormal, and the inner sign-matrix contains only $1$ or $-1$, the resulting update will satisfy:

$$||U||_{spectral} \leq ||U||_{frob} = \sqrt{nm} \quad \text{and} \quad ||U||_\infty \leq \max(\sqrt{m}, \sqrt{n}). \tag{12}$$

Additionally, instant-sign normalization provides a more fine-grained elementwise normalization than naively using the Kronecker-approximated inverse factors. As a motivating example, under an identity eigenbasis, the Shampoo update would be:

$$U = \Lambda_L^{-1/4} G \Lambda_R^{-1/4} \quad \leftrightarrow \quad u = (\lambda_L \otimes \lambda_R)^{-1/4} \, g \tag{13}$$

where notably, the diagonal component of $(\lambda_L \otimes \lambda_R)$ is not fully expressive due to being constructed out of a Kronecker product. This notion is studied in SOAP [53], who note that Shampoo (with a $1/2$ power) is equivalent to a rank-1 Adam approximation in a rotated eigenbasis. Their proposed method alleviates this issue with an additional elementwise normalization matrix for each parameter. In contrast, instant-sign normalization does not require additional parameters in memory.

As shown in Figure 2 (left), the instant-sign normalization of SPlus eliminates divergence across the board. Crucially, SPlus allows for the matrix-inversion to be cached for significantly longer intervals without collapse. Prior works on Shampoo perform recomputation every 10 steps [15, 47], and our empirical findings support that this frequency is needed to prevent divergence. In contrast, SPlus remains stable even when results are cached for over 100 steps. Utilizing this stability, SPlus can be run at a speed which outperforms Adam in reaching an equivalent validation loss not only in gradient steps, but also in *wall-clock time* (Figure 2, right).

## 4.2  To standardize learning rate across network widths, use symmetric shape-aware scaling

Learning rate is often the first hyperparameter to tune due to its outsized impact on performance. Recent works have shown that for SGD and Adam, it is possible to naturally parameterize updates such that the optimal learning rate remains constant even as network width is adjusted [54, 55]. As shown in Figure 3, neither Shampoo nor the instant-sign update above display the correct learning-rate transfer across widths. In this section, we derive a simple adjustment to introduce learning-rate transfer to our setting as well.

We start by defining a desired property, following [55] – after an update, the expected magnitude of change in *individual intermediate activations* should be invariant, regardless of network width:

$$\text{for any intermediate activation vector } x: \quad \sqrt{\frac{1}{k} \sum_{i=0}^{k} (\Delta x_i)^2} = \mathbb{O}(1). \tag{14}$$

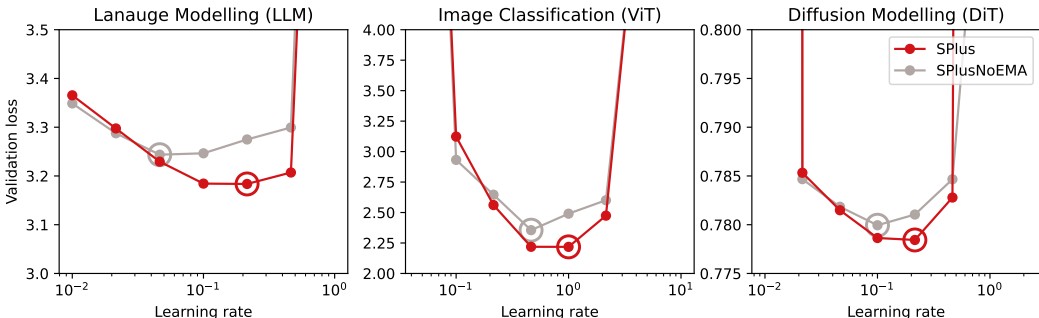

Figure 4: **Iterate averaging enables the use of higher learning rates without degradation.** Training with a higher learning rate creates a tradeoff between faster learning progress and increased parameter noise. By averaging previous iterates, parameter noise is lessened, and we can surpass the tradeoff to reveal a stronger optimal learning rate.

To achieve the above property, the norm of the update to each dense layer of shape $U = \mathbb{R}^{m \times n}$ must be properly scaled. In SPlus, Equation (12) states that the Frobenius norm of a raw instant-sign update is $\sqrt{nm}$. Thus, when considering a network with $c$-times larger width, one should divide the update by a factor of $c$. We implement this scaling without a reference width by introducing a per-layer scaling factor of $2/(m+n)$:

$$U = Q_L sign(Q_L^T G Q_R) Q_R^T \ * \ 2 \ / \ (m+n) \tag{15}$$

As shown in Figure 3, the shape-aware scaling factor enables a natural parametrization where learning rates transfer between network width. This property is especially useful for tuning, and enables a more robust default learning rate for the optimizer.

Our scaling factor is different than the "spectral" scaling of $1/m$ introduced in [54], and this is by design. We found that spectral scaling was harmful in the MLP block. Consider a $(256, 1024)$ layer and a $(1024, 256)$ layer. Under spectral scaling, the first layer would have a 4x larger per-parameter learning rate than the second. In contrast, our method is symmetric and assigns the same learning rate to both layers (as Adam does) while properly normalizing when both widths are increased. We made this design choice purely on empirical findings – our average scaling outperformed the spectral scaling in all cases, see Figure 7 – and provide a brief discussion in Appendix A.3.

### 4.3 To reduce parameter noise, make use of iterate averaging

Stochastic gradient descent methods inherently follow noisy descent directions. This noise can be broadly categorized as 1) noise from using a stochastically sub-sampled batch rather than the full dataset [34, 48], and 2) noise from linearizing the loss and taking a discrete step.

The second noise source is especially prominent in whitening-based optimizers. In naive gradient descent, the magnitude of updates will decrease as the magnitude of gradients decreases, providing a natural annealing. However, whitening-based optimizers instead utilize *normalized* updates which aim to take uniformly-sized updates. This behavior manifests in trajectories where parameters "overshoot" their ideal value, and oscillate back and forth.

While learning rate decay (and lower learning rates in general) can address these issues, they result in a tradeoff of slower learning progress. At low learning rates, learning progress is slow, and loss magnitude remains large. At high learning rates, loss is again large, but for a different reason – the presence of large noise in the parameters. Learning rate tuning can be used to locate a tradeoff between these two factors, however, is there is a way to get the best of both worlds?

We find that a more effective method of controlling the noise-progress tradeoff is via iterate averaging [39, 40, 44]. Specifically, a set of live parameters are updated with a large learning rate. A second set of slow parameters are calculated via an exponential moving average of the live parameters:

$$\theta' \leftarrow \nabla_{\theta'} L(\theta') \qquad \theta \leftarrow (1-\beta)\theta + \beta\theta'. \tag{16}$$

In this way, learning progress can remain fast, yet the effect of gradient noise is diminished, as discussed in [17] and [35]. We note that parameter averaging is a common technique in machine

| **Algorithm 1** Shampoo | **Algorithm 2** SPlus (changes in red) |
|---|---|
| **for each layer gradient** $G$ **do** | **for each layer gradient** $G$ **do** |
| $\quad G = \nabla_\theta L(\theta, x)$ where $G \in \mathbb{R}^{m \times n}$ | $\quad G = \nabla_\theta L(\theta'), x)$ where $G \in \mathbb{R}^{m \times n}$ |
| $\quad L \leftarrow (1 - \beta_2)L + \beta G G^T$ | $\quad L \leftarrow (1 - \beta_2)L + \beta G G^T$ |
| $\quad R \leftarrow (1 - \beta_2)R + \beta G^T G$ | $\quad R \leftarrow (1 - \beta_2)R + \beta G^T G$ |
| $\quad \bar{G} \leftarrow (1 - \beta_1)\bar{G} + \beta G$ | $\quad \bar{G} \leftarrow (1 - \beta_1)\bar{G} + \beta G$ |
| $\quad$ **if** $n \bmod N = 0$ **then** | $\quad$ **if** $n \bmod N = 0$ **then** |
| $\quad\quad \tilde{L}^{-1/4} \leftarrow matpow(L, -1/4)$ | $\quad\quad Q_L, \Lambda_L \leftarrow eigh(L)$ |
| $\quad\quad \tilde{R}^{-1/4} \leftarrow marpow(R, -1/4)$ | $\quad\quad Q_R, \Lambda_R \leftarrow eigh(R)$ |
| $\quad$ **end if** | $\quad$ **end if** |
| $\quad U \leftarrow \tilde{L}^{-1/4}\bar{G}\tilde{R}^{-1/4}$ | $\quad U \leftarrow Q_L sign(Q_L^T \bar{G} Q_R) Q_R^T * 2/(m+n)$ |
| $\quad \theta \leftarrow \theta + \alpha U$ | $\quad \theta' \leftarrow \theta' + \alpha U$ |
| | $\quad \theta \leftarrow (1 - \beta_3)\theta + \beta \theta'$ |
| **end for** | **end for** |

learning and has been effective in a range of domain-specific methods, e.g. image generation [19, 57], reinforcement learning [12, 51], and representation learning [14].

Figure 4 highlights the benefits of simple iterate averaging. Across the board, evaluating at the exponentially averaged parameters achieves a lower validation loss. Note that series of live parameters in the averaged and non-averaged cases are equivalent. The averaged parameters more closely reveal the "true" learning progress of utilizing a higher learning rate, which is otherwise obscured by parameter noise causing an increase in validation loss.

## 5 How does SPlus compare to prior optimizers?

We now present a thorough evaluation of SPlus alongside previous optimizers. Intentionally, we focus specifically on the Transformer architecture [52], as it has been adapted as the backbone for most large-scale neural networks today, regardless of domain or modality [10, 30, 37, 50]. Transformers are a general architecture. This flexibility means that we must be careful in evaluating their training, to avoid overfitting on a specific domain, e.g. only language modelling. To our knowledge, our setting currently represents the widest-scope evaluation of optimizers on Transformer training.

To demonstrate robustness across settings, we examine neural networks trained on three different objectives and datasets. First, we examine an autoregressive language model (**LLM**), trained on the OpenWebText [13] dataset with a sequence length of 256. Second, we examine a latent diffusion model (**DiT**) [37], trained via flow-matching [26] to generate Imagenet images encoded via a pretrained variational auto-encoder [43]. Third, we examine an image classification network (**ViT**) [10], trained to classify raw-pixel Imagenet images. All three settings are adapted directly from prior work, and utilize the same Transformer backbone.

The specific Transformer architecture is adapted from GPT-2 [8]. Layer normalization terms are applied pre-attention and pre-MLP. We remove bias terms from the network. Each objective also includes different input/output heads – a token embedding and logit predictor for language modelling, a patch embedder and patch output for diffusion modelling, and a patch embedding and class predictor for image classification. We use a momentum of $0.9$ when applicable, a linear warmup of $200$ steps followed by a constant schedule, and a weight decay of $0.1$. We train in bfloat16. See the provided code for further details.

To thoroughly compare between optimizer types, we consider the performance across different stages of training. Concretely, we construct base checkpoints by with Adam, and saving checkpoints at fixed intervals (initialization, ten thousand, and fifty thousand steps). We then evaluate each optimizer on the three checkpoints, training for ten thousand additional gradient steps. Learning rate is swept independently for each optimizer type, along a resolution of $10^{1/3}$, e.g. $(0.0001, 0.000215, 0.000464, 0.001, ...)$. Final performance is reported as validation loss after this procedure, measured on a fixed held-out validation set. The same random seed and data order are used in each run.

As loss scales vary per objective, we focus on *steps-to-Adam* as the main metric. We record the fraction of gradient steps and fraction of wallclock time required to match the performance of Adam

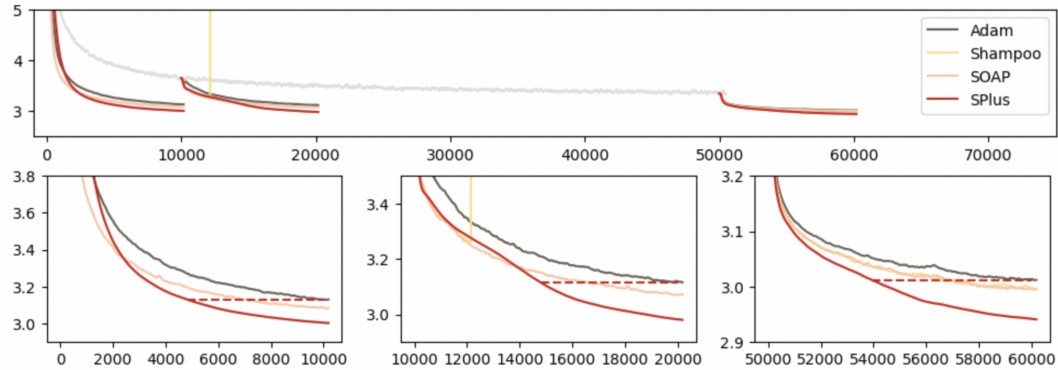

Figure 5: **Optimizers are evaluated over 10k gradient steps, starting from three distinct checkpoints per objective.** We design this setting to test robustness across objectives and across stages of training. As shown above for the LLM case, SPlus consistently reaches the same validation performance as Adam within a smaller fraction of gradient steps (dotted line).

| Method | Steps-To-Adam[1] | STA (LLM) | STA (ViT) | STA (DiT) | Time-to-Adam[1] |
|---|---|---|---|---|---|
| Naive SGD | > 10.0 | > 10.0 | > 10.0 | > 10.0 | > 10.0 |
| Adam | 1.0 | 1.0 | 1.0 | 1.0 | 1.0 |
| Sch.Free Adam | 0.679 | 0.674 | 0.698 | 0.664 | **0.654** |
| Sophia | > 1.0 | > 1.0 | n/a | n/a | > 1.0 |
| Shampoo | 0.699 [2] | 0.699 | Diverge | Diverge | 2.426 [2] |
| SOAP | 0.575 | 0.683 | 0.567 | 0.477 | 0.807 |
| PSGD | 0.652 | 0.705 | 0.615 | 0.636 | 0.940 |
| Muon | 0.832 | >1.0 | 0.920 | 0.877 | 0.934 |
| **SPlus (ours)** | **0.439** | **0.419** | **0.504** | **0.396** | **0.617** |

[1] Average values over LLM, ViT, and DiT.   [2] Only considering non-divergent settings.

Figure 6: **SPlus outperforms prior methods in both gradient steps and wallclock time, and matches Adam performance within $44 - 58\%$ of the gradient steps.** Learning rates are swept independently for each method. We examine the training of a 160M-parameter Transformer with a batch size of 1024, and a sequence/patch length of 256. Results are averaged starting from three base checkpoints, and the full results are in Table 3 of the Appendix.

on the task, measured via validation loss. Wall-clock results are machine-specific and should be seen as a rough estimate; we run all experiments on the same set of 32 TPUv3 pods, a typical run takes half a day. Results are reported utilizing the best-performing learning rate for each optimizer.

We measure performance against the following optimizers which broadly span the literature:

- **Naive SGD**, which does not modify the gradient except for scaling by a global learning rate.

- **Adam** [20], the main baseline, which keeps track of an elementwise uncentered variance, then scales the gradient elementwise.

- **Schedule-Free Adam** [9], which replaces traditional momentum with a set of live and slow parameters. Gradients are evaluated at a linear interpolation of the two.

- **Sophia** [27], which computes an elementwise estimate of the Hessian, then scales the gradient followed by a clipping step. Sophia requires auxiliary backwards passes for the Hessian calculation, and caches the results for 10 steps.

- **Shampoo** [15], as described in the background section. We do not use learning-rate grafting. Matrix inversion is performed every 10 steps (and otherwise diverges). We additionally consider Shampoo with a 1/2 power, following [36, 47].

- **SOAP** [53], a variant on Shampoo which also tracks an elementwise uncentered variance, akin to running Adam in the eigenbasis of Shampoo. Matrix inversion is performed every 100 steps.

| For frequency → | 1 | 5 | 10 | 25 | 100 | 250 | 500 |
|---|---|---|---|---|---|---|---|
| Steps-to-Adam, SPlus | 0.78 | **0.71** | 0.75 | 0.76 | 0.77 | 0.77 | 0.81 |
| Steps-to-Adam, Shampoo | 0.86 | 0.80 | 0.80 | 0.79 | Div. | Div. | Div. |
| Time per Update, SPlus | 7.39 | 1.79 | 1.09 | 0.70 | 0.48 | 0.44 | 0.43 |
| Time per Update, Shampoo | 7.41 | 1.78 | 1.09 | 0.67 | 0.42 | 0.41 | 0.40 |
| Time per Update, Adam | 0.34 | | | | | | |
| Time-to-Adam, SPlus | 16.21 | 3.34 | 2.05 | 1.20 | 0.74 | **0.65** | 0.67 |
| Time-to-Adam, Shampoo | 18.11 | 3.79 | 2.18 | 1.21 | Div. | Div. | Div |

Table 1: **Eigendecomposition frequency trades off between performance and wallclock time.** Each trial is run for 1000 gradient steps on the LLM-Init objective. Note that specific wallclock times are machine-dependent (we train on a v3-32 TPU) and may vary between systems.

| Method | LLM-Init | LLM-10K | LLM-50K | Memory Usage |
|---|---|---|---|---|
| Adam | 1.0 | 1.0 | 1.0 | $3mn$ |
| S.Free Adam | $> 1.0$ | 0.532 | 0.49 | $3mn$ |
| Adam + Iterate Averaging | 0.677 | 0.52 | 0.81 | $4mn$ |
| Shampoo | Div. | Div. | 0.699 | $2mn + 2(m^2 + n^2)$ |
| SPlus (No Averaging) | 0.729 | 0.662 | 0.685 | $2mn + 2(m^2 + n^2)$ |
| SPlus | **0.487** | **0.422** | 0.348 | $3mn + 2(m^2 + n^2)$ |
| SOAP | 0.712 | 0.66 | 0.677 | $3mn + 2(m^2 + n^2)$ |
| SOAP + Iterate Averaging | 0.513 | **0.416** | **0.317** | $4mn + 2(m^2 + n^2)$ |

Table 2: Ablating the usage of iterate averaging vs. the underlying gradient transformation.

- **PSGD** [25], which keeps track of an inverse whitening matrix calculated via iterated gradient descent rather than explicit matrix inversion. We use the Kron version which is known to perform the best.
- **Muon** [18], which performs an orthogonalization procedure on each gradient via Newton-Schulz iteration at every update, without historical information. As Muon utilizes a higher inherent learning rate, we use a weight decay of 0.001.

As displayed in Figure 6, SPlus is able to outperform prior methods across the board in both gradient steps and wallclock time. We find that a well-tuned Adam is a hard baseline to beat. For example, we were unable to match the performance of Adam with Sophia (a similar finding was reported in [58]) or with Muon for LLM training. Shampoo was especially unstable (as discussed in earlier sections), and we find that when divergence does not occur, Shampoo training curves roughly match those of SOAP, as also reported in [53]. In terms of wall-clock performance, a strong contender is Schedule-Free Adam, which does not perform any matrix-based computation and only utilizes elementwise operators. For the main table, we did not heavily tune the matrix-inversion frequency for SPlus or SOAP, and assign it a default value of 100 – Table 1 shows the effect of ablating this frequency.

## 5.1 What implementation details matter for SPlus?

We only apply the SPlus update on two-dimensional dense layers, which in the case of a Transformer, composes a majority of the backbone (the exception being LayerNorm scale parameters). We also do not apply SPlus to the domain-specific input and output layers – e.g. the token embedding, the classification head, and the convolutional patch layers. For these nonstandard parameters, we simply set the update as the sign of the momentum values. Additionally, for nonstandard parameters where the shape-dependent scaling term of Equation (15) is undefined, we use a fixed constant scaling (0.001 in our experiments). We found that this constant is not sensitive to even 10x or 0.1x pertubations, and does not need to be tuned.

We perform the above experiments over a pod of 32 TPUv3 machines, and parameters are distributed in a fully-sharded data parallel [59] setup. The SPlus update is distributed among devices. Specifically,

the per-step computations are calculated independently on each device as usual. However, for the eigendecomposition which occurs every N steps, we instead broadcast the $L$ and $R$ matrices evenly among devices. In parallel, each device then performs the eigendecomposition for its assigned matrices, then re-broadcasts the results. In this way, the most expensive step of the SPlus update can be reduced by a factor of $(1/\text{num devices})$.

To ablate the specific changes proposed in SPlus, we show in Table 2 an ablation on the interaction between iterate-averaging and instant-sign normalization, focusing specifically on the LLM setting. Iterate averaging is consistently helpful regardless of the underlying gradient transformation. This trend may be a result of our usage of a constant learning rate; as discussed in [9, 17], iterate averaging can be seen as a stand-in for learning rate decay. The downside is the memory requirement for an additional set of parameters. We note that SOAP + Iterate Averaging is competetive and can outperform SPlus. One way to contrast these methods is that the instant-sign normalization used in SPlus is a memory-efficient approximation of the internal Adam optimizer used in SOAP, which otherwise requires an additional set of parameters in memory.

## 6 Discussion and Conclusion

In this work, we present SPlus, a stable whitening optimizer for neural network training. Through a fundamentally experimental approach, we introduce three key changes to improve scalability. First, direct multiplication by the square-root inverse is replaced by instant-sign normalization, which dramatically improves stability. Second, updates are correctly scaled for learning rate to transfer among network widths. Third, iterate averaging is applied to the live parameters, which reduces parameter noise and enables using a larger learning rate.

Empirically, we show that SPlus can achieve the same validation performance as Adam with $\sim$ 44-58% of the gradient steps, and in $\sim$ 62-83% of the wall-clock time. Over a range of training objectives and checkpoints, SPlus achieves superior performance in comparison to previous optimizers.

**Code.** We provide full open-source code to replicate these experiments at github.com/kvfrans/splus.

**Limitations.** In comparison to Adam, SPlus requires storing $3nm + 2(n^2 + m^2)$ parameters per dense layer–three instances for the live parameters, slow parameters, and momentum, along with two Kronecker factors for the gradient covariances plus the cached eigenvectors. For a square matrix, this is roughly 60% more memory than Adam. Furthermore, SPlus requires nontrivial wall-clock time for matrix eigendecomposition. We note that SPlus (without iterate-averaging) uses the same amount of compute as Shampoo, and less memory than SOAP. The training settings in this work only consider Transformer architectures. While this is an intentional choice, as neural networks today have largely converged on the Transformer backbone, it remains unanswered how performance would vary on non-Transformer architectures. We utilize a constant learning rate as is common in diffusion model training [37], but is less standard in language model training.

**Future directions.** We believe that SPlus, as well as our scientific setup as a whole, opens up directions in scalable neural network optimizers. We are curious on the results of applying SPlus to large-scale training at the billion-parameter scale. By providing a thorough evaluation setup, we hope to lower the bar of experimenting with new strategies, including extensions to SPlus. For example, alternate factorizations or low-rank approximations could reduce the computational cost of whitening, and a strategy may exist in-between whitening and Hessian-based conditioning. Such ideas should be evaluated in a reproducible way, following the methodology developed here. On a practical level, we hope the efficiency and simplicity of SPlus allows the community to easily plug-and-play into their desired training objectives, and train neural networks faster as a whole.

## 7 Acknowledgments

This work was supported in part by an NSF Fellowship for KF, under grant No. DGE 2146752. Any opinions, findings, and conclusions or recommendations expressed in this material are those of the author(s) and do not necessarily reflect the views of the NSF. PA holds concurrent appointments as a Professor at UC Berkeley and as an Amazon Scholar. This paper describes work performed at UC Berkeley and is not associated with Amazon. We thank Google TPU Research Cloud (TRC) for granting us access to TPUs for research.

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

# A Appendix / supplemental material

## A.1 Pseudocode of SPlus

We provide here a snippet of the core components of SPlus, implemented in JAX. For a full implementation, check out the repo at github.com/kvfrans/splus.

```python
class SPlusState(NamedTuple):
  ema: chex.Array
  momentum: chex.Array
  sides: chex.Array
  q_sides: chex.Array
  step: int
  ema_rate: float

def splus_get_eval_params(state):
  ema_hat = jax.tree_map(lambda e: e / (1 - state.ema_rate ** state.step), state.ema)
  return ema_hat

def splus(
      learning_rate: base.ScalarOrSchedule,
      b1: float = 0.9,
      b2: float = 0.999,
      ema_rate: float = 0.999,
      eps: float = 1e-30,
      inverse_every: int = 100,
      nonstandard_constant: float = 0.001,
      weight_decay: float = 1e-2,
      mask: Optional[Union[Any, Callable[[base.Params], Any]]] = None,
      max_dim: int = 10000,
  ):

  def init_fn(params):
    momentum = otu.tree_zeros_like(params)
    ema = otu.tree_zeros_like(params)
    def sides_decomp(p):
      if len(p.shape) == 2:
        return [jnp.zeros((d, d)) if d < max_dim else None for d in p.shape]
      return None
    sides = jax.tree_map(sides_decomp, params)
    def qs_decomp(p):
      if len(p.shape) == 2:
        return [jnp.eye(d) if d < max_dim else None for d in p.shape]
    q_sides = jax.tree_map(qs_decomp, params)
    step = 0
    return SPlusState(ema, momentum, sides, q_sides, step, ema_rate)

  def update_sides(g, s):
    if len(g.shape) == 2:
      return [
        b2 * s[0] + (1 - b2) * g @ g.T if s[0] is not None else None,
        b2 * s[1] + (1 - b2) * g.T @ g if s[1] is not None else None,
      ]
    else:
      return None

  def rot(p, q):
    if len(p.shape) == 2:
      p = q[0].T @ p if q[0] is not None else p
```

```python
    p = p @ q[1] if q[1] is not None else p
  return p

def unrot(p, q):
  if len(p.shape) == 2:
    p = q[0] @ p if q[0] is not None else p
    p = p @ q[1].T if q[1] is not None else p
  return p

@jax.jit
def get_eigvecs(s):
  if s is None:
    return None
  _, q = jnp.linalg.eigh(s + eps * jnp.eye(s.shape[0]))
  return q

def update_inverse(sides):
  q_sides = jax.tree_map(get_eigvecs, sides)
  return q_sides

def update_fn(grads, state, params):
  step = state.step + 1

  # Rotate to eigenbasis, take sign, unrotate.
  momentum = jax.tree_map(lambda m, g: b1 * m + (1 - b1) * g, state.momentum, grads)
  momentum_rot = jax.tree_map(rot, momentum, state.q_sides)
  updates_rot = jax.tree_map(lambda m: jnp.sign(m), momentum_rot)
  updates = jax.tree_map(unrot, updates_rot, state.q_sides)
  sides = jax.tree_map(update_sides, grads, state.sides)
  ema = jax.tree_map(lambda e, g: ema_rate * e + (1 - ema_rate) * g, state.ema, params)

  # Every `inverse_every` steps, we update the inverse eigendecomposition.
  do_inverse = (step % inverse_every == 0) | (step == 1)
  q_sides = jax.lax.cond(do_inverse, update_inverse, lambda _ : state.q_sides, sides)

  return updates, SPlusState(ema, momentum, sides, q_sides, step, state.ema_rate)

def shape_scaling(updates, state, params):
  def shape_scale(path, u):
    path_str = '/'.join([p.key for p in path])
    if len(u.shape) == 2 and u.shape[0] < max_dim and u.shape[1] < max_dim:
      scale = (1 / (u.shape[0] + u.shape[1])/2)
    else:
      scale = nonstandard_constant
    return u * scale
  return jax.tree_util.tree_map_with_path(shape_scale, updates), None

splus_main = base.GradientTransformation(init_fn, update_fn)
splus_scaling = base.GradientTransformation(lambda _ : None, shape_scaling)
return combine.chain(
  splus_main,
  transform.add_decayed_weights(weight_decay, mask),
  transform.scale_by_learning_rate(learning_rate),
  splus_scaling
)
```

## A.2 Full results of optimizer comparisons

We present in Table 3 and Table 4 an extended table of results, comparing optimizer performance under each specific objective and starting checkpoint. As described in Section 5, all experiments are conducted using a 160M parameter Transformer model. In each plot, we compare the amount of gradient steps and/or wallclock time required to match the performance of Adam. All optimizers are trained for 10k gradient steps, and the learning rate is tuned independently. "Div." indicates that training diverges under any non-trivial learning rate. "> 1.0" indicates that at the 10k step mark, the method is unable to outperform Adam.

| Method | LLM-Init | LLM-10K | LLM-50K | ViT-Init | ViT-10K | ViT-50K | DiT-Init | DiT-10K | DiT-50K |
|---|---|---|---|---|---|---|---|---|---|
| Naive SGD | > 10.0 | > 10.0 | > 10.0 | > 10.0 | > 10.0 | > 10.0 | > 10.0 | > 10.0 | > 10.0 |
| Adam | 1.0 | 1.0 | 1.0 | 1.0 | 1.0 | 1.0 | 1.0 | 1.0 | 1.0 |
| S.Free Adam | > 1.0 | 0.532 | 0.49 | > 1.0 | 0.629 | 0.467 | > 1.0 | 0.507 | 0.487 |
| Sophia | > 1.0 | > 1.0 | > 1.0 | n/a | n/a | n/a | n/a | n/a | n/a |
| Shampoo | Div. | Div. | 0.699 | > 1.0 | > 1.0 | Div. | Div. | Div. | Div. |
| SOAP | 0.712 | 0.66 | 0.677 | 0.574 | 0.57 | 0.557 | 0.486 | 0.459 | 0.488 |
| PSGD | 0.895 | 0.628 | 0.594 | 0.535 | 0.458 | 0.852 | 0.768 | 0.412 | 0.728 |
| Muon | > 1.0 | > 1.0 | > 1.0 | 0.978 | 0.783 | > 1.0 | 0.92 | 0.878 | 0.833 |
| **SPlus** | **0.487** | **0.422** | **0.348** | **0.586** | **0.475** | **0.452** | **0.459** | **0.359** | **0.371** |

Table 3: Full results comparing steps-to-Adam.

| Method | LLM-Init | LLM-10K | LLM-50K | ViT-Init | ViT-10K | ViT-50K | DiT-Init | DiT-10K | DiT-50K |
|---|---|---|---|---|---|---|---|---|---|
| Naive SGD | > 10.0 | > 10.0 | > 10.0 | > 10.0 | > 10.0 | > 10.0 | > 10.0 | > 10.0 | > 10.0 |
| Adam | 1.0 | 1.0 | 1.0 | 1.0 | 1.0 | 1.0 | 1.0 | 1.0 | 1.0 |
| S.Free Adam | > 1.0 | **0.48** | **0.44** | > 1.0 | **0.593** | **0.441** | > 1.0 | **0.475** | **0.459** |
| Sophia | > 1.0 | > 1.0 | > 1.0 | n/a | n/a | n/a | n/a | n/a | n/a |
| Shampoo | Div. | Div. | 2.426 | > 1.0 | > 1.0 | Div. | Div. | Div. | Div. |
| SOAP | 0.951 | 0.864 | 0.886 | 0.844 | 0.811 | 0.78 | 0.734 | 0.676 | 0.72 |
| PSGD | 1.08 | 0.836 | 0.808 | 0.854 | 0.8 | 1.18 | 1.11 | 0.73 | 1.064 |
| Muon | > 1.0 | > 1.0 | > 1.0 | 0.996 | 0.79 | > 1.0 | 0.915 | 0.881 | 0.827 |
| **SPlus** | **0.651** | 0.547 | **0.447** | **0.832** | 0.674 | 0.628 | **0.707** | 0.523 | 0.545 |

Table 4: Full results comparing wallclock-to-Adam.

## A.3 Spectral scaling vs symmetric scaling

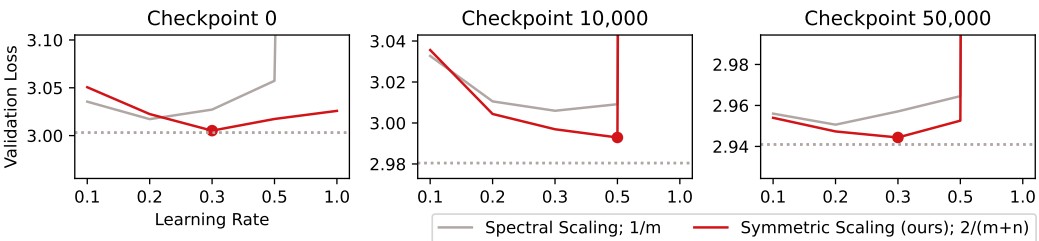

Figure 7: While both are valid strategies that enable learning rate transfer across width, we find that symmetric scaling leads to a better final performance versus spectral scaling. Dotted line shows the best-performing SPlus run *without* scaling (i.e. without learning-rate transfer properties).

In Section 4.2, we mention a difference between the SPlus symmetric scaling factor $2/(m + n)$ versus the "spectral" scaling [55] which argues for $1/m$. The spectral scaling is the correct factor such that regardless of *any dense layer input/output ratio*, the scale of activation updates remains constant. However, we find experimentally that this property is harmful for transformer training. One hypothesis is that for non-square dense layers, gradients are effectively low-rank and/or certain eigenbases do not align with incoming activation vectors.

As shown in Figure 7, the symmetric scaling that we opt for maintains the original performance, outperforming spectral scaling while allowing for learning rate transfer. The rationale behind symmetric scaling is that for the core Transformer backbones – specifically the MLP block which often consists of two layers of $(m, 4m)$ and $(4m, m)$ size – learning follows the same trajectory as if no scaling factor was used at all, i.e. a single global learning rate is applied to all parameters.

## A.4 A discussion on the Hessian, Fisher, and Empirical Fisher.

In this section, we provide a brief discussion on various distance metrics used in optimization methods. Recall that generalized gradient descent follows the steepest direction of improvement, where distance can be defined in terms of a metric matrix $M$:

$$u = \arg\min_{\Delta\theta} \underbrace{g^T \Delta\theta}_{\text{Improvement}} + \underbrace{(1/2)\Delta\theta^T M \Delta\theta}_{\text{Distance Penalty}} = M^{-1}g. \tag{17}$$

where $M$ is also referred to as a *preconditioner*.

**Hessian and Newton's method.** A particularly suitable choice for $M$ is the Hessian matrix, which is a matrix of second-order derivatives:

$$H = E_{x\sim D}\left[\nabla_\theta^2 L(\theta, x)\right]. \tag{18}$$

When the Hessian is used as a preconditioner, we arrive at Newton's method. Newton's method can be seen utilizing a *quadratic* approximation of the loss function rather than a linear one, where the penalty for taking large steps is defined entirely by the second-order effects of that step on the loss. For this reason, Newton's method is sometimes proposed as a way to avoid tuning a learning rate, and in fact for purely quadratic loss functions, Newton's method can find the global optimum in a single iteration.

Intuitively, one would desire that the Hessian is positive definite, such that the second-order term always results in a positive distance *penalty* – however, this is generally only true for convex loss functions. When the Hessian has negative eigenvalues, Newton's method can step in non-descending directions, or even diverge completely.

**Gauss-Newton matrix.** The Gauss-Newton matrix is an approximation to the Hessian using only first-order terms. For simplicity, let's assume the loss function is the mean-squared error of a single output. Denoting the network output as $f(\theta, x)$, we can expand the Hessian for a single $x, y$ pair:

$$\nabla_\theta^2 L(\theta, x) = \underbrace{\nabla_\theta f(\theta, x)\nabla_\theta f(\theta, x)^T}_{\text{Gauss-Newton term}} + \underbrace{(f_\theta(\theta, x) - y)\nabla_\theta^2 f(\theta, x)}_{\text{Dropped second-order term}}. \tag{19}$$

The Gauss-Newton approximation is often desired as it is strictly positive semi-definite, avoiding negative distance issues that the full Hessian has. Additionally, the Gauss-Newton term is simple to calculate as it only requires first-order gradients. The Gauss-Newton can be generalized to non-MSE losses [7, 21, 45] by introducing a PSD matrix $A$ between the two gradient terms:

$$G = E_{x\sim D}\left[\nabla_\theta f(\theta, x)\, A_x\, \nabla_\theta f(\theta, x)^T\right] \tag{20}$$

**Fisher information matrix and natural gradient descent.** For neural networks defining probability distributions, we can use a metric that is particular to the distribution itself:

$$F = E_{x\sim D, y\sim p_\theta(\cdot|x)}\left[\nabla_\theta \log p_\theta(y|x)\nabla_\theta \log p_\theta(y|x)^T\right] \tag{21}$$

which is known as the *Fisher information matrix*. The Fisher does *not take the loss function into account*. It is only affected by the shape of the probability distribution itself, as defined by the current neural network. When descent is performed using the Fisher as a preconditioner, it is often referred to as *natural* gradient descent [3, 49].

Natural gradient descent has the nice property that it is invariant to the parameterization of the network – to a first order, optimization will follow the same trajectory regardless of a neural networks' internal structure. This property is also true for Newton's method under affine transformations of parameters.

Note the expectation in Equation (21), which notes that $y$ must be sampled from *the current distribution*. This means that the Fisher cannot be calculated by taking the loss over samples from the dataset, and must instead use *sampled* outputs.

The Fisher can also be interpreted as the Hessian of a particular loss function, namely the expectation of log-likelihood under sampled outputs:

$$L_{\text{Fisher}} = E_{x\sim D, y\sim p_\theta(\cdot|x)}\left[\log p_\theta(y|x)\right] \tag{22}$$

where notably the second-order terms in the form of Equation (19) evaluate to zero.

**Empirical Fisher.** An approximation often used in practice is to calculate a Fisher-like matrix, but over *dataset* labels:

$$F = E_{x,y \sim D} \left[ \nabla_\theta \log p_\theta(y|x) \nabla_\theta \log p_\theta(y|x)^T \right]. \tag{23}$$

This should not be confused with the true Fisher, as studied in [34]. In fact, the empirical Fisher is closer in nature to the Gauss-Newton matrix, and is equivalent to a generalized Gauss-Newton matrix where the inner PSD matrix is constructed as $A = \nabla_f^2 \log p(y|f(\theta, x))$.

The empirical Fisher can be seen as an (uncentered) covariance of gradients. The Fisher itself is actually a *centered* covariance, since the expectation of log-likelihood gradients under sampled outputs is zero. In practice, we found that centering the empirical Fisher held no practical difference.

**Whitening metric.** The *whitening metric* [56] is the matrix square-root of the empirical Fisher:

$$W = E_{x,y \sim D} \left[ \nabla_\theta \log p_\theta(y|x) \nabla_\theta \log p_\theta(y|x)^T \right]^{1/2} \tag{24}$$

This metric is widely used in neural network training, for example, Adam is a diagonal estimate of the whitening metric, and Shampoo is a Kronecker approximation of this term. The name *whitening* refers to a property that when projected onto the whitening metric basis, the resulting preconditioned gradients have an identity covariance:

$$Cov(\hat{\nabla}, \hat{\nabla}) = I \qquad \text{where} \qquad \hat{\nabla} = W^{-1} \nabla_\theta \, L(\theta, x). \tag{25}$$

The whitening metric has the same eigenbasis as the empirical Fisher, as they are symmetric matrix powers of each other.

