# OpenReview forum: "A Stable Whitening Optimizer for Efficient Neural Network Training"
_NeurIPS.cc/2025/Conference — NeurIPS 2025 poster_

### Official Review · Reviewer_usQx · 2025-06-03

**Clarity:** 3
**Significance:** 3
**Originality:** 3
**Rating:** 4
**Confidence:** 3

**Summary:**

This paper proposes a novel whitening optimizer for neural network training that improves the convergence of learning. The major contributions include: 1. stability enhancement and computational saving for long-term matrix calculations; 2. enabling learning rate transfer across network width; 3. iterate-average learning scheme for faster learning. Numerical studies support the effectiveness of the proposed optimizer with signficant improvements on learning efficiency.

**Questions:**

1. Why is the additional wall-clock time required for SPlus negligible? (Page 9) If this is merely a numerical observation, please clarify and provide as much detail as possible for theoretical justifications.

2. Please specify how much more is the significant wall-clock cost for matrix eigen-decomposition with justifications. (Page 9) Monetary/space/time costs and their tradeoffs should be made clearer instead of only mentioning the time efficiency.

**Ethical Concerns:**

["NO or VERY MINOR ethics concerns only"]

**Final Justification:**

The complexity issue was addressed.

**Limitations:**

No. The tradeoffs among monetary/space/time costs should be better justified and compared with existing approaches for better supporting the advantages of the learning scheme and clearer statement on its disadvantages.

**Quality:**

3

**Strengths And Weaknesses:**

Strengths

1. The paper is well written.

2. Using bounded update with simple iterative-averaging scheme is novel and enhances the network stability.

3. Numerical results clearly show the superiority of the proposed optimizer over previous ones.

Weaknesses

The computational cost of the proposed optimizer should be stated and justified more clearly.

---

> ### Author Rebuttal · Authors · 2025-07-31
>
> Thank you for review. It appears the main concern is regarding the wall-clock requirements for the eigendecomposition.
>
> * **Implementation of Eigendecomposition**
>
> We first note that eigendecomposition is used in all optimizers from the Shampoo family, including SOAP and SPlus. As described in 5.1, we use a distributed method to compute this decomposition for all three methods. Regarding the claim that the additional matrix multipliation is neglible, we agree that this is misleading, and will remove this claim in the revision in favor of a more numerical analysis. See below for an additional analysis we ran investigating wall-clock time.
>
> We note that wall-clock time should be best understood as a rough estimate, as there is an inherent dependency on the hardware used, the specific distributed system, speed of inter-host communication, etc.
>
> * **Ablation on Eigendecomposition Frequency**
>
> To examine the exact relationship between wall-clock time and eigendecomposition, we ablate the frequency of the matrix eigendecomposition for both Shampoo and SPlus between [1, 5, 10, 25, 100, 250, 500]. Due to computational constraints, we measure validation loss after 1000 gradient steps (vs. 10,000 in the paper) and only on the LLM objective. All other hyperparameters are as discussed in the paper.
>
> *Validation Loss after 1K gradient steps:*
> |Frequency | 1 | 5 | 10 | 25 | 100 | 250 | 500 |
> | -- | -- | -- | -- | -- | -- | -- | -- |
> |SPlus | **3.316** | 3.332 | 3.332 | 3.335 | 3.336 | 3.336 | 3.337 |
> |Shampoo | 3.332 | 3.342 | 3.340 | 3.340 | Div. | Div. | Div. |
> | Adam | 3.36 |
>
> *Steps-to-Adam*:
> |Frequency | 1 | 5 | 10 | 25 | 100 | 250 | 500 |
> | -- | -- | -- | -- | -- | -- | -- | -- |
> |SPlus | 0.78 | **0.71** | 0.75 | 0.76 | 0.77 | 0.77 | 0.81|
> |Shampoo | 0.86 |   0.80  |   0.80 |    0.79 | Div. | Div. | Div. |
>
> As requested with **"Please specify how much more is the significant wall-clock cost for matrix eigen-decomposition with justifications"**, we make explicit the wall-clock time requrired per-step under various decomposition frequencies:
>
> *Average Time Per Update*:
> |Frequency | 1 | 5 | 10 | 25 | 100 | 250 | 500 |
> | -- | -- | -- | -- | -- | -- | -- | -- |
> |SPlus | 7.39 | 1.79 | 1.09 | 0.70 | 0.48 | 0.44 | 0.43 |
> |Shampoo | 7.41 | 1.78 | 1.09 | 0.67 | 0.42 | 0.41 | 0.40 |
> |Adam | 0.34 |
>
> *Time-to-Adam*:
> |Frequency | 1 | 5 | 10 | 25 | 100 | 250 | 500 |
> | -- | -- | -- | -- | -- | -- | -- | -- |
> |SPlus | 16.21 | 3.34 | 2.05 | 1.20 | 0.74 | **0.65** | 0.67 |
> |Shampoo | 18.11 | 3.79 | 2.18 |1.21 | Div. | Div. | Div. |
>
> Performance increases as the eigendecomposition frequency increases, but at a diminishing rate. Wallclock time scales roughly linearly with the frequency, as eigendecomposition is the main computational cost. Notably, while there is a sweet spot for the optimal frequency hyperparameter in terms of time-to-Adam (at 250 in this case), Shampoo fails to capitalize on this as training is unstable with an interval >= 100.
>
> We would like to thank you for raising relevant questions and suggestions. We believe the additional ablations have increased the clarity of the results and strengthened the paper. Please let us know if you have any remaining concerns or questions.

---

> > ### Comment · Reviewer_usQx · 2025-08-01
> >
> > Thanks for clarifying the time complexity. Is there any concern for space complexity?

---

> > > ### Author Response · Authors · 2025-08-05
> > >
> > > Regarding space complexity, please see the response to **Reviewer mSRL**, which we have reiterated here:
> > >
> > > **Memory Usage of SPlus**
> > >
> > > The additional memory requirement is a fair point and a weakness of the method in comparison to Adam. However, we note that the core SPlus update saves memory compared to the most competitive baseline (SOAP), which requires an additional set of parameters to track the second moment. In settings where iterative averaging is disabled, SPlus can roughly approximate SOAP while not requiring additional memory. We confirm this via an additional set of experiments, where SPlus is run without iterate averaging:
> > >
> > > *Validation Loss (no iterate averaging)*:
> > > | Method | LLM-Init | LLM-10K | LLM-50K | Approx Memory Usage|
> > > | -- | -- | -- | -- | -- |
> > > | Adam | 3.132 | 3.114 | 3.012 | 3nm |
> > > | SPlus No EMA | 3.087 | 3.057 | 3.010 | 2nm + 2(n^2 + m^2) |
> > > | SOAP | 3.085 | 3.072 | 2.995 | 3nm + 2(n^2 + m^2) |
> > >
> > > *Steps-to-Adam (no iterate averaging)*:
> > >  | Method | LLM-Init | LLM-10K | LLM-50K | Approx Memory Usage|
> > > | -- | -- | -- | -- | -- |
> > > | Adam | 1.0 | 1.0 | 1.0 | 3nm |
> > > | SPlus No EMA | 0.729 | 0.662 | 0.685 | 2nm + 2(n^2 + m^2) |
> > > | SOAP | 0.712 | 0.660 | 0.677 | 3nm + 2(n^2 + m^2) |
> > >
> > > In this way, one takeaway from this paper is that instant-sign normalization can be used to roughly match SOAP performance while reducing the memory footprint by one set of parameters.
> > >
> > > If the earlier and above results have strengthened the results of the paper, would you consider updating your score accordingly?

---

> > > > ### Comment · Reviewer_usQx · 2025-08-05
> > > >
> > > > Thanks for the clarification. I keep the positive score.

---

### Official Review · Reviewer_YNFF · 2025-06-20

**Clarity:** 3
**Significance:** 2
**Originality:** 1
**Rating:** 4
**Confidence:** 3

**Summary:**

This paper takes an empirical approach to study neural network optimization based on the Shampoo family of algorithms.
The authors identify three key issues with the Shampoo method and address them by proposing a method called SPlus.

The three key issues are namely that: 1) Shampoo diverges when matrix-inverses are cached for too long, 2) Shampoo updates are not properly scaled with the network width and 3) Shampoo is more sensitive to high learning rates, which are addressed by 1) performing a so-called "instant-sign normalization", 2) by introducing a scaling factor of $1/m$, where $m$ are the number of rows in the layer gradient $G \in \mathbb{R}^{m \times n}$, and finally by 3) using a iterate-averaging scheme, higher learning rates can be used.

The proposed method SPlus is compared against other preconditioned optimizers, including Adam, Sophia, Shampoo, SOAP, and Muon on Transformer training on language modeling, image classification and diffusion modeling, which highlight the proposed method over standard Adam.

**Questions:**

- I wonder whether the hypothesis of the authors for the divergence of Shampoo when the matrix-inversion is cached too long, can be empirically tested? Perhaps by checking how well the gradient and matrix-inverses are aligned throughout training?

- Can the authors also provide ablation studies for the final comparison in Steps-To-Adam and Time-to-Adam for the different modifications that they proposed? (i.e. only using the instant-normalization, only using shape-aware scaling, only using iterate-averaging, as well as
instant-normalization + iterate-averaging and instant-normalization + shape-aware scaling?) It would be sufficient for me to see it for a single task, e.g. only on the LLM or ViT.

**Ethical Concerns:**

["NO or VERY MINOR ethics concerns only"]

**Final Justification:**

This paper identifies three key limitations in Shampoo and proposes a new optimizer SPlus, which addresses all of them by introducing three modifications, namely 1) instant-normalization, 2) a width-dependent scaling factor and 3) employing iterate-averaging.

The discussion phase was more focused on understanding the effect of point 1) and 3) in SPlus through further ablation studies, and its comparison to other methods from the family of "whitening optimizers", including Muon, SOAP and Shampoo.

I suggest acceptance by giving a score of 4 because the proposed method SPlus does address important limitations in Shampoo, further expanding the family of "whitening optimizers". The reason for not giving a higher score is because the proposed modifications are not that novel and mainly taken from other optimization methods (e.g. instant-sign normalization resembles Signum and the benefit of iterate-averaging was already known for SGD.)

**Limitations:**

yes.

**Paper Formatting Concerns:**

/

**Quality:**

3

**Strengths And Weaknesses:**

Strength:
- The paper is clearly written and easy to follow.
- The improvement through the suggested modifications are illustrated well and together they are able to outperform Adam (and many other optimizers) both in terms of iterations and clock-time across all tested settings.
- The authors provide Code, details about the experiments as well as Pseudocode of SPlus in the appendix.

Weaknesses:
- The proposed modifications are not necessarily novel per se. Instant-sign normalization is essentially Signum [1] in Shampoo's eigenbasis, which is reminiscent of SOAP being a rank-1 Adam approximation in a rotated eigenbasis. Also iterate-averaging has already been quite known [2]. As such the I am not sure about the novelty of this work. (The authors also refer to previous work, this is not my critique point.)
- I think the comparison of SPlus to Adam is overselling the proposed method a bit. If one checks Figure 6, then many of recently proposed methods are quite competitive to SPlus. For instance Schedule-Free Adam is only 6% slower than SPlus. Given that Schedule-Free Adam improves considerably on Adam, I wonder how a Schedule-Free version of SOAP, PSGD and Muon would fare.
- In general I am always a bit skeptical about wall-clock time runs, as it also highly depends on how each of the methods are implemented in Code, which have the risk to obscure insights. For instance the authors describe how the updates are computed in SPlus in lines 270-276 and how this is optimized for SPlus (which is a great thing to mention), but it is less clear how other methods are implemented and used.
- The ablation studies clearly show the effect and benefit of each modification to Shampoo.

[1] Jeremy Bernstein, Yu-Xiang Wang, Kamyar Azizzadenesheli, and Animashree Anandkumar. 320 signsgd: Compressed optimisation for non-convex problems. In International Conference on 321 Machine Learning, pp. 560–569. PMLR, 2018
[2] Pavel Izmailov, Dmitrii Podoprikhin, Timur Garipov, Dmitry Vetrov, and Andrew Gordon 348 Wilson. Averaging weights leads to wider optima and better generalization. arXiv preprint 349 arXiv:1803.05407, 2018.

---

> ### Author Rebuttal · Authors · 2025-07-31
>
> Thank you for your detailed review and questions. Please see our response below:
>
> * **Ablating the proposed changes**
>
> Addressing the request for an ablation of each modification, we ran an additional set of experiments to properly ablate the changes in SPlus, specifically focusing on the interaction between iterate-averaging and instant-sign normalization in the LLM setting. We specifically ran an experiment of SPlus without the use of iterate averaging. We did not ablate shape-aware scaling, as this requires another sweep to find the optimal learning rate, which we are unable to perform due to compute limitations.
>
> To address the request for **"I wonder how a Schedule-Free version of SOAP, PSGD and Muon would fare"**, we additionally compare to SOAP + iterate averaging, as SOAP is the highest-performing baseline method in our experiments. (EMA iterate averaging was more performant than Schedule-Free in our experiments)
>
> *Steps-to-Adam:*
> | Method | LLM-Init | LLM-10K | LLM-50K |
> | -- | -- | -- | -- |
> | Adam  |  1.0 | 1.0 | 1.0 |
> | Shampoo  | Div.  | Div. | 0.699 |
> | Shampoo + Instant-Sign (i.e. SPlus without EMA) | 0.729  | 0.662 | 0.685 |
> | Shampoo + Instant-Sign + Iterate Averaging (SPlus) | 0.487 | 0.422 | 0.348 |
> | SOAP | 0.712 | 0.660 | 0.677 |
> | SOAP + Iterate Averaging | 0.513 | 0.416 | 0.317 |
>
> As shown in the above table, the usage of instant-sign normalization is important to stabilize Shampoo. Both SPlus and SOAP benefit from iterate averaging. SOAP performs slightly better at later checkpoints, but requires keeping an additional set of parameters in memory to track the second moment.
>
> * **Wall-clock time and inversion frequency**
>
> Thank you for mentioning the skepticism on wall-clock item -- we agree that wall-clock times are hard to standardize and should be seen as a rough estimate --  there is an inherent dependency on the hardware used, the specific distributed system, speed of inter-host communication, etc. The methods with the longest time-per-step in our experiment suite are Shampoo/SOAP/SPlus, especially at frequent inversion times. For all three of these methods, we use the same distributed behavior.
>
> To understand the tradeoff between wall-clock time and performance, we conduct an ablation on the matrix-inversion frequency in Shampoo and SPlus between [1, 5, 10, 25, 100, 250, 500]. Due to computational constraints, we measure validation loss after 1000 gradient steps (vs. 10,000 in the paper) and only on the LLM objective. All other hyperparameters are as discussed in the paper.
>
> *Validation Loss after 1K gradient steps:*
> |Frequency | 1 | 5 | 10 | 25 | 100 | 250 | 500 |
> | -- | -- | -- | -- | -- | -- | -- | -- |
> |SPlus | **3.316** | 3.332 | 3.332 | 3.335 | 3.336 | 3.336 | 3.337 |
> |Shampoo | 3.332 | 3.342 | 3.340 | 3.340 | Div. | Div. | Div. |
> | Adam | 3.36 |
>
> *Steps-to-Adam*:
> |Frequency | 1 | 5 | 10 | 25 | 100 | 250 | 500 |
> | -- | -- | -- | -- | -- | -- | -- | -- |
> |SPlus | 0.78 | **0.71** | 0.75 | 0.76 | 0.77 | 0.77 | 0.81|
> |Shampoo | 0.86 |   0.80  |   0.80 |    0.79 | Div. | Div. | Div. |
>
> *Average Time Per Update*:
> |Frequency | 1 | 5 | 10 | 25 | 100 | 250 | 500 |
> | -- | -- | -- | -- | -- | -- | -- | -- |
> |SPlus | 7.39 | 1.79 | 1.09 | 0.70 | 0.48 | 0.44 | 0.43 |
> |Shampoo | 7.41 | 1.78 | 1.09 | 0.67 | 0.42 | 0.41 | 0.40 |
> |Adam | 0.34 |
>
> *Time-to-Adam*:
> |Frequency | 1 | 5 | 10 | 25 | 100 | 250 | 500 |
> | -- | -- | -- | -- | -- | -- | -- | -- |
> |SPlus | 16.21 | 3.34 | 2.05 | 1.20 | 0.74 | **0.65** | 0.67 |
> |Shampoo | 18.11 | 3.79 | 2.18 |1.21 | Div. | Div. | Div. |
>
> Performance increases as the eigendecomposition frequency increases, but at a diminishing rate. Wallclock time scales roughly linearly with the frequency, as eigendecomposition is the main computational cost. Notably, while there is a sweet spot for the optimal frequency hyperparameter in terms of time-to-Adam (at 250 in this case), Shampoo fails to capitalize on this as training is unstable with an interval >= 100.

---

> ### Comment · Reviewer_YNFF · 2025-08-02
>
> Thank you for the detailed responses and providing additional ablation studies.
>
> Most of my questions have been addressed, but I do have some remaining ones and follow-up questions:
>  Looking at the ablation study, it appears that instant-normalization has a stabilizing, but almost no acceleration effect as Shampoo with and without instant-norm have a very similar value at a later checkpoint, while the speed-up mainly comes from iterate averaging, which is also evident by the fact that SOAP + iterate-averaging performs very similarly to SPlus if not better.
> I wonder if iterate-averaging alone already provides some stabilizing effect as well. Have the authors considered running SPlus without instant-normalization?
>
> I have also read your rebuttal to **Reviewer Xpwz** and noticed that Muon with smaller weight decay is much more competitive (without iterate-averaging). In particular due to smaller per-iteration complexity it converges faster in wall-clock time. I wonder if Muon can achieve even better performance by adding iterate-averaging?

---

> ### Author Response · Authors · 2025-08-05
>
> Thank you for your response! Regarding SPlus without instant-sign normalization -- this would essentially be Shampoo with iterate averaging, and as shown in the above table, Shampoo has a tendency to diverge when the eigenbases are cached for long periods. Iterate averaging would not prevent divergence, as the diverged parameters explode in magnitude and are unusable even when averaged.
>
> Regarding Muon, yes, Muon can likely achieve a similar gain in performance from iterate averaging as SPlus/SOAP do. As shown in the table you are referring to, Muon is competitive with SPlus/SOAP in many cases, and has an efficient implementation in terms of wall-clock time -- however, in the above results Muon tends to perform worse when training from later checkpoints, especially on the DiT and ViT cases which have a higher variance batch. (Diffusion has inherently high variance due to the noised objective, and ViT has less signal per-batch due to the single classification target, in comparison to language modelling which calculates a loss at each token). This can be intuitively understood, as Muon approximates the orthogonalization per-batch, whereas SPlus/SOAP/Shampoo can utilize historical information.
>
> We hope these answers your questions. As mentioned in the original paper, the famiIy of 'whitening' optimizers (e.g. Shampoo, SOAP, PSGD, Muon, SPlus) are similar in form. One takeaway from this paper is that in the Shampoo family, SPlus provides a way to roughly match the performance of SOAP while reducing the memory footprint. In all methods, iterate averaging can increase performance at each step (at the cost of additional memory, which we aim to save via the previous technique in SPlus).
>
> If the additional experiments have strengthened your view on the paper, would you consider updating the score accordingly?

---

> ### Comment · Reviewer_YNFF · 2025-08-05
>
> Thank you for the clarifications!
>
> I would like to thank the authors for taking the time to clarify my questions and providing detailed explanations. I got a better understanding of SPlus and its performance compared to other optimizers in the family of "whitening" optimizers as a whole now.
>
> I acknowledge the contributions as valuable and suggest acceptance of this work and keep my positive score.

---

### Official Review · Reviewer_Xpwz · 2025-06-28

**Clarity:** 3
**Significance:** 3
**Originality:** 2
**Rating:** 4
**Confidence:** 4

**Summary:**

This paper introduces SPlus, a stable and efficient optimizer that addresses three key limitations of the Shampoo algorithm: susceptibility to divergence, shape sensitivity across network widths, and elevated gradient noise under large learning rates. To mitigate these issues, the authors propose three targeted modifications. Extensive evaluations on Transformer architectures across three representative tasks demonstrate that SPlus yields substantial improvements in both convergence speed and wall-clock efficiency.

**Questions:**

1. The effect of the averaging coefficient $\beta_3$ in parameter averaging is unclear. It would be helpful to provide empirical sensitivity analysis or practical guidance on choosing $\beta_3$ .
2. Can a fixed choice of the hyperparameter $\beta_3$  maintain an appropriate update magnitude across different training phases? Specifically, does the parameter averaging mechanism risk inducing insufficient updates in the early stages, or causing instability in later training due to oscillatory behavior?
3. Could additional theoretical or intuitive justification be provided for why parameter averaging helps mitigate optimization noise?

**Ethical Concerns:**

["NO or VERY MINOR ethics concerns only"]

**Final Justification:**

The proposed method is effecitve and the writing is good, and my concerns are well addressed in rebuttal. I keep positive with "Borderline accept"

**Limitations:**

Yes

**Quality:**

3

**Strengths And Weaknesses:**

**Strengths**

1.The paper proposes targeted solutions to common optimization challenges such as instability and sensitivity to hyper-parameters of Shampoo algorithm, making approximately second-order optimizer potentially applicable to train deep neural networks.

2.The experiments cover multiple training stages and diverse tasks (LLM, ViT, DiT), with comparisons against a strong set of baselines. SPlus shows clear improvements in both convergence and wall-clock efficiency.

3.The paper is well-written and easy to follow. Source code is provided, supporting strong reproducibility.



**Weaknesses**

**Limited Theoretical Depth:** The proposed improvements are primarily empirical in nature, targeting issues such as instability and gradient noise during optimization. While effective, techniques like instant-sign normalization and parameter averaging build on existing methods, and the paper lacks deeper theoretical analysis—e.g., why parameter averaging effectively mitigates noise in this setting.

**Increased Memory Overhead:** As noted in the limitations, the method incurs additional memory cost compared to Adam, due to the need to maintain historical parameters for averaging, which may limit its applicability in resource-constrained environments. It is better for this paper provide the comparison of wall-clock time cost.

**Limited Architectural and Scale Generalization:** The generalization of SPlus to larger-scale models and non-Transformer architectures remains untested. The steps-to-Adam metric is not sufficient to evaluate the performance. It is better to provide the final results (e.g., Accuracy on ImageNet classification and others for training LLMs) when using different optimizer with equal/sufficient training steps.  Besides, It is not clear   how the introduced hyper-parameters $\beta_3$ affects the performance. I think this paper should provide the ablation study for hyper-parameters $\beta_3$ of SPlus.



Other minors:

(1) The notation is not self-contained, e.g, What is the notation $g, \Delta \theta$ meaning? This paper should provide the illustration for self-containing.  Similar problems also happens in Eqn. 17, what is $\theta^{'}$? It seems has something wrong in the left formulation  of Eqn.17

 (2)The axes in Figures 3 and 5 lack clear labels.

---

> ### Author Rebuttal · Authors · 2025-07-30
>
> Thank you for your review and detailed questions. Please see the detailed response:
>
> * **Ablating iterate averaging $\beta_3$**
>
> To answer the question on the sensitivity of the $\beta_3$ parameter, we ran an additional set of ablations, focusing on the LLM setting. Our experimental setup already measures the performance at three stages of training (initialization, 10k checkpoint, 50k checkpoint), and we report values ablating $\beta_3$ for SPlus:
>
> *Validation Loss of SPlus*:
> | Starting Checkpoint | $\beta_3 = 0$ | $\beta_3 = 0.9$ | $\beta_3 = 0.99$ | $\beta_3 = 0.995$ | **$\beta_3 = 0.999$** | $\beta_3 = 0.9999$ |
> | -- | -- | -- | -- | -- | -- | -- |
> | LLM at Init  | 3.085 | 3.064 | 3.028 | 3.017 | **3.003** | 3.077 |
> | LLM at 10K  | 3.062 | 3.047 | 3.012 | 3.002 | **2.988** | 3.303 |
> | LLM at 50K | 3.010 | 2.995 | 2.964 | 2.954 | **2.936** | 3.035 |
>
> *Steps-to-Adam of SPlus*:
> | Starting Checkpoint | $\beta_3 = 0$ | $\beta_3 = 0.9$ | $\beta_3 = 0.99$ | $\beta_3 = 0.995$ | **$\beta_3 = 0.999$** | $\beta_3 = 0.9999$ |
> | -- | -- | -- | -- | -- | -- | -- |
> | LLM at Init  | 0.73 | 0.616 | 0.479 | **0.461** | 0.486 | 0.771 |
> | LLM at 10K  | 0.667 | 0.556 | 0.409 | **0.384** | 0.419 | Div. |
> | LLM at 50K | 0.998 | 0.792 | 0.495 | 0.425 | **0.383** | Div. |
>
> As shown in the above table, we can get slightly better performance by tuning the $\beta_3$ parameter per training setting. However, using a default value of 0.999 (as we do in the main paper) allows for nearly the same performance across the board, regardless of the starting checkpoint. The intuitive reason for the effectiveness of iterate averaging, especially under a constant learning rate, can be viewed as reducing the effect of noise during optimization -- if we assume that each gradient steps introduces some independent noise due to linearization/stochastic batches, then iterate averaging enables the noise to cancel out (variance of the mean of i.i.d. random variables decays linearly with the sample size). We will include this discussion in the next revision.
>
>
> * **Memory Usage of SPlus**
>
> The additional memory requirement is a fair point and a weakness of the method. However, we note that the core SPlus update *saves* memory compared to the most competitive baseline (SOAP), which requires an *additional* set of parameters to track the second moment. In settings where iterative averaging is disabled, SPlus can roughly approximate SOAP while not requiring additional memory. We confirm this via an additional set of experiments, where SPlus is run without iterate averaging:
>
> *Validation Loss (no iterate averaging)*:
> | Method | LLM-Init | LLM-10K | LLM-50K | Approx Memory Usage|
> | -- | -- | -- | -- | -- |
> | Adam  | 3.132  | 3.114 | 3.012 | 3nm |
> | SPlus No EMA  | 3.087  | 3.057 | 3.010 | 2nm + 2(n^2 + m^2) |
> | SOAP | 3.085 | 3.072 | 2.995 | 3nm + 2(n^2 + m^2) |
>
> *Steps-to-Adam (no iterate averaging)*:
> | Method | LLM-Init | LLM-10K | LLM-50K | Approx Memory Usage|
> | -- | -- | -- | -- | -- |
> | Adam  | 1.0  | 1.0 | 1.0 | 3nm |
> | SPlus No EMA  | 0.729  | 0.662 | 0.685 | 2nm + 2(n^2 + m^2) |
> | SOAP | 0.712 | 0.660 | 0.677 | 3nm + 2(n^2 + m^2) |
>
> * **Notation and figures**
>
> Regarding the notation and the axes, thank you for catching these issues, and we have fixed them in the next revision. $g$ in our case represents the gradient with relation to the loss function (i.e. $\nabla L(\theta) $), and $\Delta \theta$ is an arbitrary vector to take the argmin over. $\theta’$ is a set of "live" parameters, which are averaged into $\theta$.
>
> * **Why do we use the Steps-to-Adam metric?**:
>
> We chose steps-to-Adam as it is an interpretable way to understand the effective speedup of one optimizer to the rest, and many prior works report a speedup over Adam, e.g [1, 2]. Raw validation losses are less interpretable, as the relative difference is harder to compare. That said, in the above tables, we have also provided raw validation losses.
>
> * **Regarding the Transformer-only experiments**
>
> It is a fair point that our experiments limit only to Transformers, as mentioned in the limitations section. We note that our experimental setup is still larger than many previous works [1, 2] that only consider language modelling -- we additionally consider diffusion modelling and image classification.
>
> We would like to thank you for raising relevant questions and suggestions. We believe the additional ablations have increased the clarity of the results and strengthened the paper. Please let us know if you have any remaining concerns or questions.
>
> [1] Sophia: A Scalable Stochastic Second-order Optimizer for Language Model Pre-training. Lui et al, 2023.
>
> [2] SOAP: Improving and Stabilizing Shampoo using Adam. Vyas et al, 2024.

---

> > ### Comment · Reviewer_Xpwz · 2025-08-05
> > **Comments on responses**
> >
> > Dear authors:
> >
> > Thanks for the clarification. My main concerns are addressed, and I keep the positive score.

---

### Official Review · Reviewer_oPR8 · 2025-06-29

**Clarity:** 3
**Significance:** 3
**Originality:** 3
**Rating:** 4
**Confidence:** 4

**Summary:**

This paper proposes a new optimizer termed SPlus.  The algorithm is positioned as an improvement over shampoo.  SPlus maintains exponential moving averages of the left and right squared gradients, and then normalizes the new gradient by conducting sign-normalization in the shampoo eigenbasis, which is re-computed every so often.  The authors say that SPlus allows for these eigenvalue computations to be performed less regularly than in shampoo, which yields wall clock gains.  The main justification is empirical effectiveness.  SPlus also includes two other components: a scaling adjustment which allows the learning rate to be transferred across widths (as in Mu-P), and iterate averaging, which averages out the parameter noise/oscillations and is claimed to allow larger learning rates.

**Questions:**

If I understand things correctly, if we put aside iterate averaging and shape-aware scaling, then SPlus is very similar to Muon.  In particular, SPlus with $\beta_2 = 0$ is almost exactly Muon (or rather, an idealized version of Muon which exactly computes the matrix sign rather than using the Newton-Schultz iteration), with the only remaining differences being: whether L/R is computed from $G$ (SPlus) or $\overline{G}$ (Muon); and (2) whether momentum is parameterized as $\overline{G} \leftarrow \(1 - \beta_1) \overline{G} + \beta_1 G$ (SPlus) vs $\overline{G} \leftarrow \beta_1 \overline{G} + G$ (Muon), which are the same up to a rescaling of the learning rate by $1 / (1 - \beta_1)$.  I am trying to understand which of the differences between SPlus and Muon are responsible for SPlus's superior performance that was observed in experiments.  I have the following questions:

 - If you remove iterate averaging and shape-aware scaling from SPlus, or alternatively if you add them to Muon, how do the two algorithms compare?  I am especially thinking about iterate averaging.
- How does SPlus with $\beta_2 = 0$ compare to SPlus with $\beta_2 = 0.9$ (i.e the value used in your experiments)?

**Ethical Concerns:**

["NO or VERY MINOR ethics concerns only"]

**Limitations:**

Yes

**Quality:**

2

**Strengths And Weaknesses:**

Strengths:
 - the experiments (all on transformers) show that SPlus beats not only Adam but also several newer methods including Muon, SOAP, and PSGD.

Weaknesses:
 - I have questions about the comparison to Muon  -- see below
 - There are a number of mathematical typos:
   - equation 1, which is the first equation of the paper, is wrong; the RHS should be $-(1/2\alpha) g$, not $\alpha g$
   - equation 3 is also wrong; the RHS should be $- \frac{1}{2} M^{-1} g$
   - equation 11, which is the main equation describing the proposed algorithm, is wrong; the transpose should be on the second $Q_L$ and not the first one
   - I believe the SPlus algorithm box is missing $2$ subscripts for some of the $\beta_2$'s.

---

> ### Author Rebuttal · Authors · 2025-07-30
>
> Thank you for the detailed review and analysis. Please find our detailed response below:
>
> * **Relation and comparison to Muon**
>
> Regarding the relation to Muon, we also agree with your analysis here. The relationship between Shampoo and Muon is studied in [1], and SPlus can be understood as an intermediate point where the eigenvalues are calculated via the sign operator, but the eigenvectors are calculated over the historical average. We will further clarify this relationship in the paper.
>
> Regarding the gap in performance, we found through further analysis that the original Muon reported numbers were using a default weight decay that caused a substantial degradation. In particular, we use weight decay = 0.1*lr for all experiments -- however, this setting is suboptimal for Muon which has an inherently high learning rate. For a more representative comparison, we re-ran the Muon experiments with a weight decay of 0.001 * lr, and report the new numbers:
>
> *Steps-to-Adam:*
> | Method | LLM-Init | LLM-10K | LLM-50K | ViT-Init | ViT-10K | ViT-50K | DiT-Init | DiT-10K | DiT-50K |
> |---|---|---|---|---|---|---|---|---|---|
> SOAP |  0.712 | 0.66 | 0.677 | 0.574 | 0.57 | 0.557 |0.486 | 0.459 | 0.488 |
> Muon (new) |0.636|0.68|0.91|0.567|0.652|1.0+|0.556|0.507|0.507|
> SPlus |  0.487 | 0.422 | 0.348 | 0.586 | 0.475 | 0.452 | 0.459 | 0.359 | 0.371
>
> *Time-to-Adam:*
> | Method | LLM-Init | LLM-10K | LLM-50K | ViT-Init | ViT-10K | ViT-50K | DiT-Init | DiT-10K | DiT-50K |
> |---|---|---|---|---|---|---|---|---|---|
> SOAP |  0.951 | 0.864 | 0.886 | 0.844 | 0.811 | 0.78 | 0.734 | 0.676 | 0.72 |
> Muon (new) |0.611|0.66|0.89|0.618|0.698|1.0+|0.57|0.515|0.59|
> SPlus |  0.651 | 0.547 | 0.447 | 0.832 | 0.674 | 0.628 | 0.707 | 0.523 | 0.545 |
>
> As shown above, SOAP and SPlus tend to reach the desired validation loss in less gradient steps than Muon. However, Muon can reach this point in less wall-clock time, likely due to its efficient use of Newton-Schulz iteration versus explicit eigendecomposition.
>
> * **Ablating the Effect of Iterate Averaging**
>
> We ran an additional experiment ablating the effect of iterate averaging (via EMA) on SPlus, focusing specifically on the LLM setting due to compute limitations. We also compare to introducing iterate averaging to SOAP, as SOAP is the most competitive baseline in terms of Steps-to-Adam.
>
> *Validation Loss*:
> | Method | LLM-Init | LLM-10K | LLM-50K |
> | -- | -- | -- | -- |
> | Adam  | 3.132  | 3.114 | 3.012 |
> | SPlus No EMA  | 3.087  | 3.057 | 3.010 |
> | SOAP | 3.085 | 3.072 | 2.995 |
> | Muon | 3.061 | 3.061 | 3.007 |
> | SPlus  | 3.003  | 2.978 | 2.936 |
> | SOAP + EMA | 2.998| 2.993 | 2.931 |
>
> *Steps-to-Adam:*
> | Method | LLM-Init | LLM-10K | LLM-50K |
> | -- | -- | -- | -- |
> | Adam  |  1.0 | 1.0 | 1.0 |
> | SPlus No EMA  | 0.729  | 0.662 | 0.685 |
> | SOAP | 0.712 | 0.660 | 0.677 |
> | Muon | 0.636| 0.68| 0.91 |
> | SPlus  | 0.487 | 0.422 | 0.348 |
> | SOAP + EMA | 0.513 | 0.416 | 0.317 |
>
> SPlus without EMA performs similarly to SOAP (and Muon), and all three can be understood as approximating a particular spectral whitening objective (see Section 3, Eq. 4). In our specific setting with a constant learning rate, iterative averaging provides an additional performance gain. The practical benefit of SPlus over SOAP is that SPlus requires one less set of parameters to be kept in memory.
>
> Regarding mathematical typos, thank you for pointing these out, and they have been fixed for the next revision.
>
> We would like to thank you for raising relevant questions and suggestions. We believe the additional ablations have increased the clarity of the results and strengthened the paper. Please let us know if you have any remaining concerns or questions.
>
>
> [1] Jeremy Bernstein and Laker Newhouse. Old optimizer, new norm: An anthology. 2024.

---

> > ### Comment · Reviewer_oPR8 · 2025-08-04
> > **response**
> >
> > Thanks for running these experiments!
> >
> > To clarify - would it be possible for SPlus to use Newton-Schultz iteration instead of explicit eigendecomposition?

---

> > > ### Author Response · Authors · 2025-08-05
> > >
> > > Great question -- it would not be possible naively, as the eigendecomposition gives us the (historical) left/right eigenvectors explicitly, while Newton-Shulz implicitly solves for the orthogonal matrix UV^T. There is no simple way to extract U or V from the combined matrix UV^T. If one tried to run Newton-Shulz on the GG^T matrices, you would get the identity! (as GG^T is full-rank and symmetrical)
> > >
> > > ```
> > > import numpy as np
> > >
> > > g = np.random.rand(4, 20)
> > > ggt = g @ g.T
> > > print(ggt.shape)
> > > print(ggt)
> > >
> > > coeffs = (3, -16/5, 6/5) # Many choices work here.
> > > def newton_schulz_iterator(x):
> > >     a = x @ x.T
> > >     b = coeffs[1] * a + coeffs[2] * a @ a
> > >     return coeffs[0] * x + b @ x
> > > x = ggt
> > > x /= np.linalg.norm(x) + 1e-6  # Singular values must be in [-1, 1]
> > > for _ in range(5):  # Iterate a few times to converge.
> > >     x = newton_schulz_iterator(x)
> > > print('Newton-Schulz orthogonalization\n', x)
> > >
> > > # Explicit orthogonalization
> > > u, s, vh = np.linalg.svd(ggt)
> > > print('Explicit orthogonalization\n', u @ vh)
> > > ```
> > >
> > > ```
> > > (4, 4)
> > > [[4.52943642 4.34923569 4.21630621 4.21347935]
> > >  [4.34923569 8.74947178 5.75485439 6.18474024]
> > >  [4.21630621 5.75485439 7.15446298 5.41502261]
> > >  [4.21347935 6.18474024 5.41502261 7.6006473 ]]
> > > Newton-Schulz orthogonalization
> > >  [[ 9.96821455e-01 -3.39008106e-04  2.19201580e-03  1.15344933e-03]
> > >  [-3.39008106e-04  9.97126618e-01 -4.52450965e-03  8.55999996e-03]
> > >  [ 2.19201580e-03 -4.52450965e-03  9.95145616e-01  8.73224205e-03]
> > >  [ 1.15344933e-03  8.55999996e-03  8.73224205e-03  9.82407800e-01]]
> > > Explicit orthogonalization
> > >  [[ 1.00000000e+00  2.56983382e-16  1.97278891e-18  1.85961893e-16]
> > >  [ 2.06052557e-16  1.00000000e+00 -5.51062184e-18  1.44590563e-16]
> > >  [ 1.56194126e-16  1.20230648e-16  1.00000000e+00  7.58146768e-17]
> > >  [-1.00514801e-16  1.84793521e-16  2.05541700e-16  1.00000000e+00]]
> > > ```
> > >
> > > That said, there may be promising ways to combine Newton-Schulz decomposition with the *historic* gradients even beyond the current momentum term (as is done in Muon), however, we believe this is beyond the scope of this paper.
> > >
> > > We believe the above experiments strengthened the paper. If you agree, would you consider updating the score accordingly?

---

### Official Review · Reviewer_mSRL · 2025-07-03

**Clarity:** 3
**Significance:** 3
**Originality:** 3
**Rating:** 4
**Confidence:** 4

**Summary:**

This paper introduces SPlus, a novel optimizer designed to stabilize and accelerate training for large neural networks, particularly Transformers. It builds upon the Shampoo family of whitening-based optimizers, which estimate a preconditioning matrix via Kronecker-factored approximations of gradient covariances.

The authors identify three critical weaknesses in existing Shampoo-like optimizers:
- Instability from stale matrix inverses.
- Learning rate scaling issues across network widths.
- Parameter noise at high learning rates.
To address these, SPlus introduces:
- Instant-sign normalization in place of magnitude-based scaling, ensuring bounded updates and eliminating the divergence observed in Shampoo.
- Shape-aware scaling to ensure that optimal learning rates can transfer across different model widths.
- Iterate averaging to mitigate parameter noise while maintaining fast learning dynamics.

Extensive experiments on Transformer architectures (language modeling, image classification, diffusion modeling) show that SPlus matches the validation loss of Adam in ~44% fewer gradient steps and in ~62% of the wall-clock time.

**Questions:**

One paper I think is very related with the proposed method is NGPlus [1], which is a second-order method and saves computational cost compared with KFAC and Shampoo.

[1] Yang, Minghan, et al. "An efficient Fisher matrix approximation method for large-scale neural network optimization." IEEE Transactions on Pattern Analysis and Machine Intelligence 45.5 (2022): 5391-5403.

Did you experiment with low-rank approximations for L and R? For example, truncating eigen decompositions might reduce both memory and compute costs. Do you have empirical results or intuition on how such approximations might affect stability or convergence speed?

Have you measured wall-clock speed on significantly larger models (e.g. billion-parameter LLMs)? The eigen decompositions could become disproportionately costly.

**Ethical Concerns:**

["NO or VERY MINOR ethics concerns only"]

**Limitations:**

yes

**Quality:**

4

**Strengths And Weaknesses:**

Strengths

The paper convincingly demonstrates how cached matrix inverses can destabilize training in whitening-based optimizers. The explanation of eigen-decomposition and sign-normalization provides strong intuition for SPlus’ improvements.


Adapting insights from width-scaling theory to whitening optimizers is a significant practical contribution. This should reduce the trial-and-error process for hyperparameter tuning as models scale.

Evaluations are thorough, covering different tasks and various training stages. SPlus consistently outperforms other methods, including Shampoo, SOAP, PSGD, and Sophia.


Weaknesses

All experiments focus on Transformer-based models. It is unclear whether the same improvements would hold for other architectures such as CNNs, RNNs, or graph neural networks.

SPlus increases memory usage significantly (∼60% more than Adam). For very large models (e.g. billion-parameter LLMs), this could become a practical barrier. While the authors mention low-rank factorization as future work, no experiments or ablations are presented to quantify potential savings.

SPlus performs eigendecomposition every N steps (defaulting to 100). The paper does not fully explore how sensitive performance is to this frequency, especially in larger models where eigen decompositions can become bottlenecks.

---

> ### Author Rebuttal · Authors · 2025-07-30
>
> Thank you for the detailed review and suggestions. Please see our detailed response and additional experiments below:
>
> * **How does eigendecompisition frequency affect performance and wallclock time?**
>
> This is a great question, and we ran an additional set of experiments to answer it. In the following experiments, we run SPlus and Shampoo over a range of eigendecomposition frequencies between [1, 5, 10, 25, 100, 250, 500]. Due to computational constraints, we measure validation loss after 1000 gradient steps (vs. 10,000 in the paper) and only on the LLM objective. All other hyperparameters are as discussed in the paper.
>
> *Validation Loss after 1K gradient steps:*
> |Frequency | 1 | 5 | 10 | 25 | 100 | 250 | 500 |
> | -- | -- | -- | -- | -- | -- | -- | -- |
> |SPlus | **3.316** | 3.332 | 3.332 | 3.335 | 3.336 | 3.336 | 3.337 |
> |Shampoo | 3.332 | 3.342 | 3.340 | 3.340 | Div. | Div. | Div. |
> | Adam | 3.36 |
>
> *Steps-to-Adam*:
> |Frequency | 1 | 5 | 10 | 25 | 100 | 250 | 500 |
> | -- | -- | -- | -- | -- | -- | -- | -- |
> |SPlus | 0.78 | **0.71** | 0.75 | 0.76 | 0.77 | 0.77 | 0.81|
> |Shampoo | 0.86 |   0.80  |   0.80 |    0.79 | Div. | Div. | Div. |
>
> *Average Time Per Update*:
> |Frequency | 1 | 5 | 10 | 25 | 100 | 250 | 500 |
> | -- | -- | -- | -- | -- | -- | -- | -- |
> |SPlus | 7.39 | 1.79 | 1.09 | 0.70 | 0.48 | 0.44 | 0.43 |
> |Shampoo | 7.41 | 1.78 | 1.09 | 0.67 | 0.42 | 0.41 | 0.40 |
> |Adam | 0.34 |
>
> *Time-to-Adam*:
> |Frequency | 1 | 5 | 10 | 25 | 100 | 250 | 500 |
> | -- | -- | -- | -- | -- | -- | -- | -- |
> |SPlus | 16.21 | 3.34 | 2.05 | 1.20 | 0.74 | **0.65** | 0.67 |
> |Shampoo | 18.11 | 3.79 | 2.18 |1.21 | Div. | Div. | Div. |
>
> Performance increases as the eigendecomposition frequency increases, but at a diminishing rate. Wallclock time scales roughly linearly with the frequency, as eigendecomposition is the main computational cost. Notably, while there is a sweet spot for the optimal frequency hyperparameter in terms of time-to-Adam (at 250 in this case), Shampoo fails to capitalize on this as training is unstable with an interval >= 100.
>
> We note that wall-clock time should be best understood as a rough estimate, as there is an inherent dependency on the hardware used, the specific distributed system, speed of inter-host communication, etc.
>
> * **Memory Usage of SPlus**
>
> The additional memory requirement is a fair point and a weakness of the method. However, we note that the core SPlus update *saves* memory compared to the most competitive baseline (SOAP), which requires an *additional* set of parameters to track the second moment. In settings where iterative averaging is disabled, SPlus can roughly approximate SOAP while not requiring additional memory. We confirm this via an additional set of experiments, where SPlus is run without iterate averaging:
>
> *Validation Loss (no iterate averaging)*:
> | Method | LLM-Init | LLM-10K | LLM-50K | Approx Memory Usage|
> | -- | -- | -- | -- | -- |
> | Adam  | 3.132  | 3.114 | 3.012 | 3nm |
> | SPlus No EMA  | 3.087  | 3.057 | 3.010 | 2nm + 2(n^2 + m^2) |
> | SOAP | 3.085 | 3.072 | 2.995 | 3nm + 2(n^2 + m^2) |
>
> *Steps-to-Adam (no iterate averaging)*:
> | Method | LLM-Init | LLM-10K | LLM-50K | Approx Memory Usage|
> | -- | -- | -- | -- | -- |
> | Adam  | 1.0  | 1.0 | 1.0 | 3nm |
> | SPlus No EMA  | 0.729  | 0.662 | 0.685 | 2nm + 2(n^2 + m^2) |
> | SOAP | 0.712 | 0.660 | 0.677 | 3nm + 2(n^2 + m^2) |
>
> * **Potential methods of improving compute efficiency**
>
> As noted in the future work section, we agree that there is a fruitful research direction on methods which can improve the computational efficiency of SPlus and other Shampoo-like methods. We did not conduct specific experiments on algorithmic approximations to the eigendecomposition (e.g. low-rank approximations) as we view them as beyond the scope of this paper.  In terms of wall-clock time for billion-parameter LLMs, while we are also curious on this result, we are unable to properly experiment with such large-scale models due to compute limitations. Thank you for the reference to NGPlus! We will update the paper to introduce a citation to this, and describe potential techniques to reduce the computational footprint.
>
> We would like to thank you for raising relevant questions and suggestions. We believe the additional ablations have increased the clarity of the results and strengthened the paper. Please let us know if you have any remaining concerns or questions.

---

### Note · Authors · 2025-08-15

Dear AC and reviewers,

Thank you for the feedback and discussion, along with the uniformly positive reviews. We believe that during the response period we've addressed the majority of questions raised by reviewers, including:
- Clarification on the memory usage of SPlus. We note that our core update *saves memory compared to the strongest baseline method (SOAP)*, which utilizes an extra set of parameters to track the second moment.
- Additional experiments ablating the effect of matrix inversion frequency. The per-gradient performance of SPlus can be improved with more frequent preconditioning (we use 100 in the paper), however, this is at the cost of wall-clock time.
- Additional experiments ablating the usage of iterate averaging and the usage of instant-sign normalization vs. the naive Shampoo update.

Accordingly, we plan to update a revision to the paper to include:
- A nuanced breakdown of the performance gain (and computational cost) of each modification we propose in the paper (instant-sign normalization, shape-aware scaling, and iterate averaging).
- The results of the above ablations regarding inversion frequency and the choice of beta3.
- Fixes to mathematical typos and missing subscripts.


We also note that we have improved the Muon baseline (see response to Reviewer oPR8), having discovered that the default weight decay we use in the main paper (0.1) was suboptimal specifically with the Muon optimizer -- we have re-run these results with a weight decay of 0.001, and believe the new numbers lead to a more accurate comparison. We believe the main takeaways of the paper remain unchanged.

---

### Decision · Program_Chairs · 2025-09-17

**Decision:**

Accept (poster)

**Comment:**

The paper introduces SPlus, an optimizer designed to improve the stability and acceleration of training large neural networks and especially transformers. It addresses three key weaknesses of existing Shampoo-like optimizers: instability from stale matrix inverses, learning rate scaling issues across network widths, and parameter noise at high learning rates. SPlus proposes instant-sign normalization for bounded updates, shape-aware scaling for optimal learning rate transfer, and iterate averaging to mitigate parameter noise. Through extensive experiments on transformer architectures, SPlus demonstrates significant improvements in convergence speed and wall-clock time over Adam.

A positive point of the paper is how it demonstrates how cached matrix inverses destabilize training in whitening-based optimizers and SPlus' improvements through eigen-decomposition and sign-normalization. Adapting width-scaling theory insights to whitening optimizer is a significant practical contribution that should reduce hyperparameter tuning effort. Thorough evaluations across different tasks and training stages show SPlus' consistent outperformance of various methods (including Shampoo, SOAP, PSGD, and Sophia). The paper is clearly written, easy to follow, and additionally provides source code.

Despite its strengths, several concerns were raised.
* There is limited architectural generalization, as all experiments focus on transformers.  (And while low-rank factorization is mentioned as future work, no experiments quantify potential savings)
* The increased memory usage of SPlus (∼60% more than Adam), which could be a practical barrier.
* There is doubt over the quality of the muon baseline (but the authors have/will address this in revision)
* The lack of exploration into the sensitivity of performance to the frequency of eigendecomposition, which can become a bottleneck in larger models.
* There is a lack deeper theoretical analysis for some of the proposed modifications, such as why parameter averaging effectively mitigates noise.

The reviewers explicitly found some minor corrections, which should be addressed, and the authors have promised a list of changes.

(Accept)